# Position: Peer Review Should Be Calibrated via LLM Scoring

**Zijin Chen** [* 1 2]  **Lesui Yu** [* 1]  **Xiaofei Liao** [1]  **Hai Jin** [1]  **Qinbin Li** [† 1]

## Abstract

As submission volumes grow, AI conference peer review increasingly suffers from scale drift and non-comparable scoring: similar rationales can yield markedly different numeric ratings due to subjective calibration and occasional incoherent or strategic scoring, even though scores often strongly influence outcomes. This position paper argues that **AI conference workflows should incorporate an LLM-driven calibration layer that maps reviewer rationales (e.g., strengths and weaknesses) into consistent and auditable anchor scores.** The residual between a reviewer's reported score and the anchor score turns rationale–score misalignment into a measurable signal for targeted escalation. We instantiate an end-to-end pipeline and apply it to OpenReview data from ICLR 2023–2025, using leakage-aware analyses and post-cutoff datasets to quantify severity/leniency patterns and where misalignment concentrates. We further propose a lightweight post-check—requesting added justification or score revision when residuals are large—and estimate its impact via an offline counterfactual simulation. Finally, we outline an adoption playbook and governance boundaries, emphasizing that the LLM audits scoring coherence rather than replacing human judgment or making accept/reject decisions.

## 1. Introduction

AI conferences are experiencing an unprecedented scaling crisis in peer review. Submission counts continue to rise, while qualified reviewer capacity grows much more

---
[*]Equal contribution [1]National Engineering Research Center for Big Data Technology and System, Services Computing Technology and System Lab, Cluster and Grid Computing Lab, School of Computer Science and Technology, Huazhong University of Science and Technology, Wuhan, Hubei, China [2]Beijing Jiaotong University. Correspondence to: Qinbin Li <qinbin@hust.edu.cn>.

*Proceedings of the 43rd International Conference on Machine Learning*, Seoul, South Korea. PMLR 306, 2026. Copyright 2026 by the author(s).

slowly, pushing the system toward a high-load regime where reviewers face heavier assignments and tighter deadlines. This trend has been explicitly documented in large-venue analyses and interventions, and in empirical studies of OpenReview-based conferences that report increasing noise and perceived arbitrariness (Stelmakh et al., 2021; Tran et al., 2020; Sculley et al., 2018; Beygelzimer et al., 2023). As the field grows, the community's dissatisfaction has become increasingly visible: authors and area chairs routinely report inconsistent reviews, questionable scores, and "lottery-like" outcomes. When review quality degrades, it erodes trust in conferences as gatekeepers for scientific credit and career progression.

A seemingly natural response is to automate review with *large language models* (LLMs). In practice, LLMs are already used to draft reviews, summarize papers, or generate pros/cons, and recent evidence suggests a measurable fraction of peer-review text in major AI venues after ChatGPT may have been substantially LLM-modified beyond minor edits (Liang et al., 2024a; Zhuang et al., 2025). At first glance, full AI reviewing sounds attractive: if human reviewers are overloaded and inconsistent, why not let a model enforce a single standard? However, fully replacing human judgment with AI judgment is not desirable. Scientific evaluation is not just surface-level correctness; it requires tacit expertise, evolving norms, and value judgments about what directions are promising. More importantly, once authors believe that an automated model is a primary decision-maker, scientific writing can drift toward model-targeted optimization—optimizing for a model's preferences, heuristics, and blind spots rather than for clarity, rigor, and long-term impact. Recent empirical studies that evaluate LLMs as autonomous reviewers highlight substantial misalignment risks and systematic biases in score predictions, reinforcing the case that LLMs should not be treated as replacements for reviewers (Akella et al., 2025).

Instead of automating judgment, we focus on a narrower and more actionable failure mode that is widely blamed for review frustration: **unreliable scoring**. In most AI venues, reviewers must compress multi-dimensional strengths and weaknesses into a single scalar score that heavily influences decisions. Figure 1 illustrates how strongly acceptance probability correlates with the average score in practice; near the decision boundary, small score shifts can flip outcomes. Yet

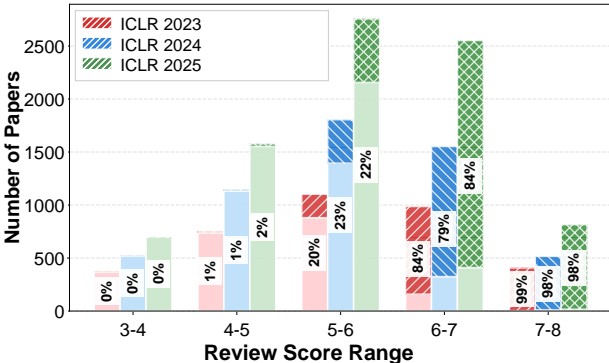

*Figure 1.* Acceptance rate by average review score for ICLR 2023–2025

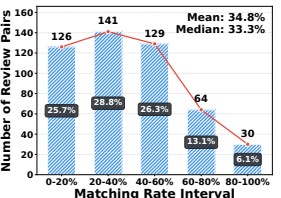 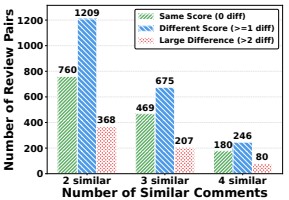

*(a)* Rationale overlap among same-score review pairs for the same paper

*(b)* Score disagreement among reviews with aligned comments for the same paper

*Figure 2.* Evidence of scoring fragility

the mapping from review text to score is under-specified and highly subjective. Two reviewers may articulate similar critiques but assign very different scores because their internal baselines and severity thresholds differ. For example, one reviewer may treat a missing ablation as a minor revision request, while another views it as a decisive rejection reason. Rater effects further amplify this mismatch: junior reviewers may score more conservatively, close-to-topic experts may penalize missing baselines more harshly, and senior reviewers may calibrate differently. These differences are not inherently wrong—diverse perspectives in the review text are valuable. The problem is that a single scalar score forces these perspectives onto incomparable scales, injecting avoidable randomness into a process where scores act as a hard filter. Consistent with this concern, large-scale consistency experiments show substantial disagreement in accept/reject outcomes under independent reviewer assignments (Cortes & Lawrence, 2021; Beygelzimer et al., 2023).

**This position paper argues for an LLM-based calibration layer for scores in AI conference peer review.** Crucially, we do not advocate LLMs as replacement reviewers. We advocate a strict division of labor: humans provide scientific judgment and qualitative rationale, while LLMs provide a standardized, auditable mapping from the written rationale to a score under a fixed rubric.

Concretely, given (i) a reviewer's written strengths and weaknesses and (optionally) (ii) the paper content, an LLM produces an *anchor score* under a fixed prompt and rubric. By comparing this anchor score with the reviewer's reported score, we quantify rationale–score misalignment. Large misalignments trigger a lightweight feedback loop: the reviewer is asked to either (i) provide additional justification that supports the score, or (ii) revise the score to better align with the written rationale. Importantly, the LLM does not introduce new criticisms or make accept/reject decisions; it only enforces a more consistent *interface* between what the reviewer has already said and the implication of the numeric score.

We instantiate this proposal with an end-to-end calibration pipeline and study OpenReview data from ICLR 2023–2025. We quantify the prevalence and structure of rationale–score deviations, evaluate leakage-aware variants using older-model reruns and post-cutoff datasets, and study how calibration could be integrated into existing workflows. Our results suggest that LLM-assisted calibration can reduce avoidable variance and improve consistency while preserving the human reviewer as the sole source of scientific judgment.

## 2. Problem: Why Human Scores Are a Weak Interface

Numeric ratings are widely used as a coordination signal in AI conference peer review. However, when a single scalar score is treated as a decision interface, it becomes fragile and hard to interpret across reviewers. Our empirical analysis highlights two failure modes: (i) a score is a low-bandwidth, under-specified compression of rich textual rationales, and (ii) reviewers apply different internal scoring scales, leading to non-comparable ratings even when their stated viewpoints overlap. These failures matter because AI conference workflows often treat small score differences as decisive signals for paper acceptance.

### 2.1. Low Bandwidth and Under-Specification of Human Scores

Mapping a written review to a single score is an under-specified and lossy encoding. In principle, similar qualitative observations should lead to similar numeric adjustments, but a scalar cannot preserve the trade-offs and uncertainty expressed in text. As a result, materially different stances can map to the same point on the score scale.

To quantify this ambiguity, we analyze 490 pairs of ICLR 2023 reviews that evaluate the same paper and assign the same overall score. We compute semantic similarity using an SBERT model (Reimers & Gurevych, 2019) and perform point-by-point matching as follows: for each comment (pros or cons) in review A, we find its most similar counterpart in review B via cosine similarity; if the similarity exceeds a

threshold, we count it as a match. The matching rate is the proportion of comments in review A that find a semantic match in review B. Figure 2a shows that same-score review pairs share limited overlap in concrete viewpoints: the mean matching rate is 34.8% (median 33.3%), and 25.7% of pairs fall below 20%. This pattern indicates that a single score often corresponds to multiple distinct rationale profiles, making the score an ambiguous summary of what the reviewer actually emphasized.

This observation aligns with Geertz's distinction between thick and thin description (Geertz, 1993). Written reviews provide context, caveats, and trade-offs, whereas a scalar rating discards much of that structure. In practice, this compression makes it difficult to rely on scores alone to infer whether a paper is viewed as high potential with key weaknesses, or merely adequate but unexciting. In short, same scores do not imply comparable rationales.

### 2.2. Scale Drift and Reviewer-Specific Miscalibration

A second weakness is that scores are often non-comparable across reviewers. Even when two reviewers highlight similar strengths and weaknesses, they may apply different internal baselines. For the same issue, such as missing ablations, one reviewer may treat it as a decisive flaw while another treats it as a minor revision item. These differences in severity thresholds create scale drift that is separate from substantive disagreement about the paper.

We test for scale drift by focusing on review pairs for the same paper that exhibit overlap in specific comments. For each ICLR 2023 paper, we embed all pros and cons with SBERT and identify pairs of reviews where at least two comments are semantically similar, indicating partial alignment in what the reviewers noticed. Despite this overlap, Figure 2b shows substantial score disagreement: across 3,734 such review pairs from 1,958 papers, 60.1% assign different scores (mean difference 1.31; median 1.0), and 18.2% differ by more than 2 points. These gaps are large enough to matter near typical decision boundaries. The key point is that similar rationale content does not guarantee comparable scores, which is consistent with reviewer-specific calibration differences.

A plausible mechanism is psychological anchoring (Tversky & Kahneman, 1974). In the absence of a shared external standard, reviewers may form idiosyncratic numeric baselines from prior experiences and local norms. Related work argues that individual rating scales can be arbitrary and difficult to align (Ammar & Shah, 2012). Taken together, our evidence suggests that scores provide a weak interface: they are both information-poor summaries of text and non-comparable across reviewers. This motivates a calibration layer that audits whether the score is coherent with the stated rationales under a shared rubric.

## 3. Proposed Mechanism: LLM-Based Calibration for Paper Scoring

### 3.1. Mechanism Overview

Our position is that conference workflows should treat the numeric rating as an auditable interface rather than a reviewer-specific convention. Accordingly, we propose an LLM-driven calibration layer that standardizes how a reviewer's written rationale implies a score while preserving human scientific judgment. As shown in Figure 3, the calibration layer takes as input (i) the paper content and (ii) each first-round review's stated strengths and weaknesses (plus the reported overall rating), produces an *anchor score*, and computes a residual between the anchor and the reported score. The residual turns rationale–score misalignment into a measurable signal that can trigger selective follow-ups.

This design follows three principles commonly emphasized in human-centered AI practice: assist rather than replace human decision-makers (Shneiderman, 2020), intervene minimally in existing workflows (Amershi et al., 2019), and keep outputs inspectable and accountable via structured artifacts (Mitchell et al., 2019; Gebru et al., 2021). Concretely, the LLM is not allowed to introduce new critiques or make accept/reject decisions; it only audits coherence between what the reviewer writes and what the numeric score implies.

### 3.2. Context-Aware Anchor Scoring and Residual Signals

A core design choice is that anchor scoring is context-aware: the LLM conditions not only on the rationale text, but also on the paper content. This is intentional. The same surface-level issue can reasonably carry different weight depending on the paper (e.g., "missing ablations" is more severe for a claim of state-of-the-art performance than for a conceptual position paper; "limited baselines" matters differently across subfields and task maturity). Similarly, reviewers may express the same point with different certainty or severity (e.g., "minor" vs. "fatal"). Our goal is therefore not to force a global lookup table where each rationale maps to a fixed point adjustment. Instead, we require that the mapping procedure be consistent: under a rubric and prompting protocol, the LLM should translate the reviewer's stated rationale in the context of the given paper into a score that is stable and auditable.

Operationally, for each paper $p$ and reviewer $j$, we (optionally) normalize the review text into itemized rationale statements and then compute an anchor score $s_{p,j}^A$ by asking the LLM to assign rubric-grounded contributions to each stated strength and weakness conditional on the paper. Thus, two reviews that mention superficially similar points can legitimately receive different anchor adjustments if (i) the paper's claims and evidence differ, or (ii) the reviewer's language

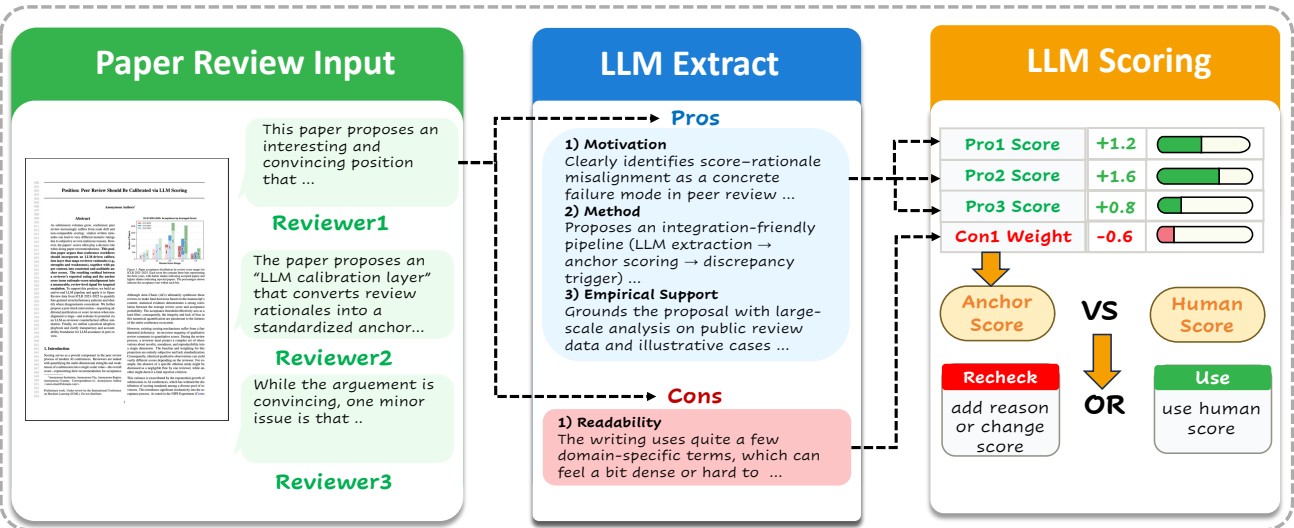

*Figure 3.* LLM-based calibration layer for review scoring

indicates different severity. What we enforce is **procedural consistency**: with identical inputs (paper + rationale items) and identical decoding settings, the anchor score should be reproducible (validated empirically in Section 4.1).

Let $s_{p,j}^H \in \{1, \ldots, 10\}$ denote the reviewer's reported rating. We define the residual $r_{p,j} = s_{p,j}^H - s_{p,j}^A$. Large $|r_{p,j}|$ indicates that the reported rating is difficult to justify from the stated rationale given the paper under the fixed rubric, and therefore serves as a triage flag. At the paper level, we aggregate residuals across reviewers to quantify non-comparable scoring (e.g., dispersion or span) and to prioritize cases where clarification is most valuable. Implementation details (including the rationale itemization step, scoring prompts, and safeguards) are provided in Appendices A and A.3.

### 3.3. Residual-Triggered Post-Check

The calibration layer is designed to be selective: when $|r_{p,j}| > \tau$, the system triggers a lightweight post-check that asks the reviewer to reconcile the mismatch by either (i) adding additional paper-grounded justification that supports the original rating, or (ii) revising the rating to better align with the written rationale. Otherwise, the system takes no action and the original human score remains unchanged. This keeps the intervention sparse and preserves legitimate substantive disagreement for area-chair adjudication.

## 4. Empirical Evidence

**Overview.** This section evaluates our central claim that rationale–score misalignment is a measurable and workflow-relevant failure mode in large-scale peer review, and that an LLM-based calibration layer can surface it as an auditable signal. We organize the evidence around five ques-

tions: (i) whether anchor scoring is stable enough for auditing, (ii) whether non-comparable scoring is substantial and increasing, (iii) whether extreme disagreements have interpretable structure, (iv) whether residual-triggered post-checks can plausibly reconcile mismatches, and (v) whether anchor scores provide useful signals for borderline papers. To address leakage and model-family concerns, we report contamination controls, post-cutoff datasets, prompt sensitivity checks, and cross-model stress tests alongside the main analyses. Unless otherwise noted, we use first-round reviews only, and paper-level statistics are computed on papers with at least two reviews from ICLR 2023–2025. We release the code repository on `https://github.com/wutaghost/LLMscore-ICLR-OpenReview`.

### 4.1. Evidence 1: Anchor Scoring Is Reproducible and Responsive

*Evidence 1: Anchor scores are reproducible under fixed inputs and respond predictably when reviewer rationales are perturbed, supporting their use as an auditable measurement reference.*

**Setup.** Residual-based auditing only makes sense if the anchor score is stable when inputs are held fixed; otherwise residuals would confound reviewer inconsistency with model generation noise. We therefore measure test–retest reliability by sampling roughly 20% of submissions per year and running the anchor scoring pipeline three times per review under identical inputs and decoding settings. This yields 2,604 (ICLR 2023), 2,709 (ICLR 2024), and 8,827 (ICLR 2025) first-round reviews scored by the anchor pipeline. We evaluate stability using (i) run-to-run variance of anchor scores and (ii) cross-run correlation (Ap-

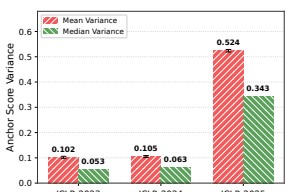

*(a)* Fixed-protocol test–retest reliability

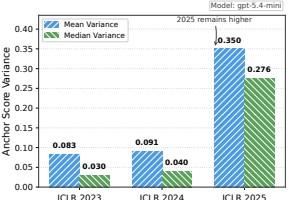

*(b)* Variance control with gpt-5.4-mini

*Figure 4.* Anchor-score reproducibility checks

pendices A.3 and B provide the pipeline and test–retest details). As a measurement sanity check, we also hold the paper and reviewer score fixed while perturbing the reviewer rationale text, including supportive, generic, and less supportive variants (Table 2 in Appendix).

**Findings.** Anchor scoring is highly reproducible across all years (Figure 4), behaving near-deterministically under the fixed protocol. While 2025 exhibits higher variability than earlier years—consistent with broader heterogeneity at larger scale—the magnitude remains small relative to the 1–10 rating scale. The same variance pattern remains when the analysis is repeated with gpt-5.4-mini, a model whose pretraining explicitly includes ICLR 2023–2025, indicating that the 2025 increase is not simply a leaked-versus-non-leaked transition. This supports treating large residuals in later evidence as systematic rationale–score misalignment rather than artifacts of stochastic scoring. The rationale perturbation check behaves in the expected direction: more supportive rationales reduce the mean absolute residual to 0.641; original rationales yield 1.333; generic boilerplate yields 1.863; and less supportive rationales increase it to 2.573. This check does not constitute causal inference about peer-review outcomes; it verifies that the anchor is sensitive to the audited text-to-score interface rather than invariant to reviewer rationale content.

### 4.2. Evidence 2: Residual Dispersion Reveals Growing Non-Comparable Scoring

*Evidence 2: Paper-level residual dispersion is substantial and increases from ICLR 2023 to 2025, indicating growing non-comparable scoring among reviewers of the same paper.*

**Setup.** We quantify how differently reviewers translate written rationales into numeric ratings by aggregating residuals within each paper. For each paper $p$, we compute dispersion over residuals $\{r_{p,j}\}_{j \in \mathcal{R}(p)}$. Our primary metric is **bias span** (max residual minus min residual), which captures the maximum calibration disagreement between any two reviewers of the same paper; we also report within-paper residual standard deviation as a complementary measure.

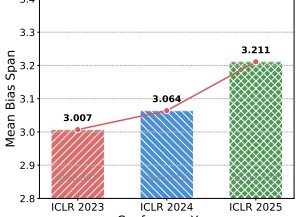

*(a)* Mean bias span by conference year

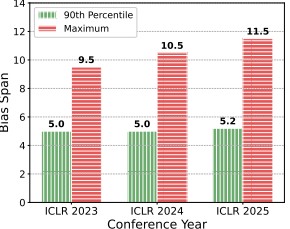

*(b)* Tail behavior in extreme cases

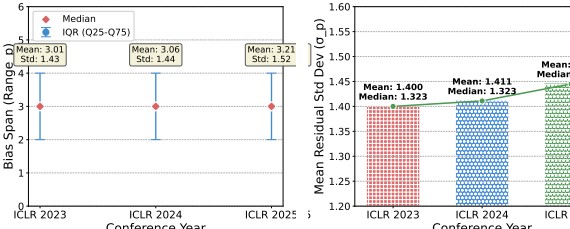

*(c)* Bias-span distribution summary

*(d)* Within-paper residual dispersion

*Figure 5.* Residual dispersion across ICLR 2023–2025

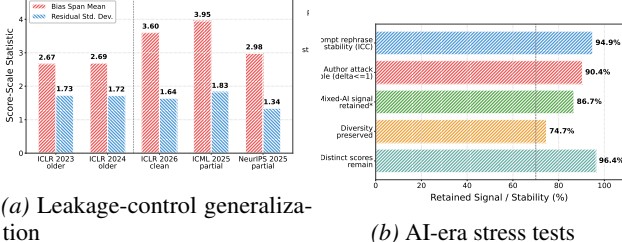

*(a)* Leakage-control generalization

*(b)* AI-era stress tests

*Figure 6.* Robustness checks for residual-based calibration

We compute these paper-level statistics for each year separately and summarize their distributions at the conference level (Appendix A.3.3 includes additional derived metrics).

**Findings.** Residual dispersion is large in absolute terms and grows over time (Figure 5). The mean bias span increases from 2023 to 2025, and the tail becomes heavier in the latest cycle. A large bias span indicates that even after conditioning on the same auditing reference, reviewers can differ by multiple points in how they map similar rationales to final ratings. This is the regime where calibration is most useful: it makes scale disagreement explicit and routable for clarification near the boundary. The pattern also survives leakage-aware variants. With `gpt-3.5-turbo-1106` on ICLR 2023–2024, residual dispersion remains substantial; with `gpt-4o-mini` on post-cutoff ICLR 2026, ICML 2025, and NeurIPS 2025, bias-span means remain high at 3.601, 3.950, and 2.980, with residual standard deviations of 1.643, 1.826, and 1.337. These results (Figure 6) reduce the likelihood that the dispersion pattern is a simple memorization artifact.

**Robustness checks.** Figure 6 also summarizes four AI-era robustness checks that constrain how the anchor should be interpreted. First, semantically equivalent prompt rephrasings produce stable scores ($ICC(2, 1) = 0.9493$), while explicitly strict or lenient prompts shift outputs; this supports prompt pinning for a full review cycle. Second, author-side LLM rewriting of paper style leaves anchor scores largely stable when the original human pros/cons are fixed (Pearson $r = 0.9016$, with 90.4% of changes within one point), indicating that the signal is driven mainly by reviewer rationales rather than surface writing style. Third, mixed human/LLM reviewer simulations retain substantial residual dispersion because uncoordinated reviewer-side tools introduce heterogeneous mappings. Fourth, diversity-preservation analysis shows that 96.4% of semantically different reviewer pairs still retain distinct scores after calibration, so the procedure does not collapse all disagreement into one score. Together, these checks support treating the anchor as an audit artifact tied to a specific reviewer rationale, model version, prompt, rubric, and paper version, rather than as an alternative paper-quality label.

### 4.3. Evidence 3: Disagreement Driven by Evidence Thresholds and High-Stakes Innovation

*Evidence 3: Extreme disagreement is structurally interpretable, primarily driven by disputes over **evidence strength** and **novelty**. While presentation issues are frequent, high-weight disagreements on innovation drive the largest residual spans.*

**Setup.** To determine whether large residual dispersion reflects arbitrary noise or structural disagreement, we inspect papers in the top $\sim$1% by bias span ($N = 219$ total across three years). We map extracted rationale items to a post-hoc taxonomy and analyze their **quantized weights** (Appendix D). This approach allows us to distinguish between arguments that are merely *frequent* versus those with sufficient *magnitude* to cause extreme scoring divergence.

**Findings.** The anatomy of disagreement is structured rather than uniformly random (Figure 7). **Evidence Strength** remains the dominant source of friction ($\sim$30% of arguments), suggesting calibration failures often stem from subjective thresholds for empirical sufficiency. Crucially, while **Clarity** is frequently cited, its contribution to score divergence is limited by lower weights ($\sim$0.8). In contrast, **Novelty/Innovation** can act as a high-leverage multiplier: though less frequent ($\sim$8%), it commands the highest positive weight impact (+1.50). Consequently, the most extreme bias spans typically reflect fundamental splits on contribution significance—where one reviewer rewards innovation while another strictly penalizes evidence gaps—rather than simple confusion over presentation.

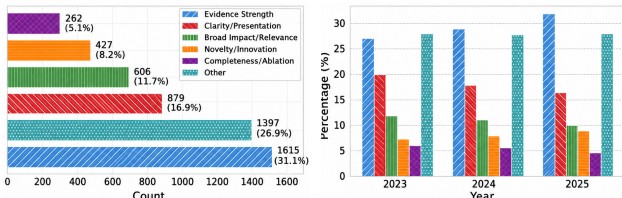

*(a)* Overall rationale-category distribution

*(b)* Category distribution by year

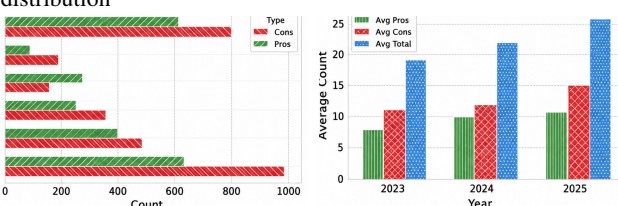

*(c)* Rationale types split by pros and cons

*(d)* Average pros and cons per paper

*Figure 7.* Rationale taxonomy and weight impact for extreme-disagreement papers

### 4.4. Evidence 4: Residual-Triggered Post-Check Is a Plausible Corrective Mechanism

*Evidence 4: In an offline counterfactual proxy setting, a residual-triggered post-check often reconciles mismatches and reduces residual magnitude.*

**Setup.** We evaluate whether residuals can support a workflow-compatible corrective intervention. Because counterfactual human reviewer behavior is unobservable in retrospective datasets, we test mechanistic plausibility via an offline LLM-as-reviewer simulation on $N = 200$ triggered cases. A post-check is triggered when $|r_{p,j}| > \tau$: the reviewer is shown the rationale items and anchor score and asked to reconcile the mismatch by either (i) revising the score to better align with the stated rationale or (ii) keeping the score while providing additional, paper-grounded justification. We measure (a) reduction in $|r_{p,j}|$ after reconciliation and (b) how often reconciliation occurs via score revision versus added justification (details in Appendix E).

**Findings.** The post-check reduces residual magnitude in simulation and frequently resolves cases via score revision (Figure 8). This suggests that residual-triggered escalation can plausibly correct rationale–score mismatches while intervening only on a small subset of reviews. The effect is similar when the simulated reviewer is replaced with Qwen-2.5-72B-Instruct while the anchor model is held fixed (1.778 versus 1.874 points of residual reduction), reducing the chance that the result is only a same-model artifact. Because simulation may overestimate compliance under real incentives, this evidence should be interpreted as feasibility support rather than a claim about true human response rates, motivating the human-in-the-loop pilot discussed in

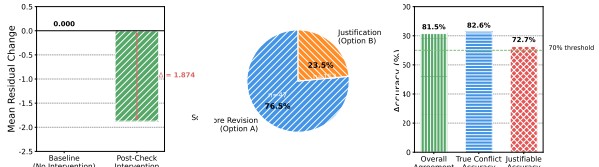

*(a)* Residual reduc-*(b)* Reviewer choice *(c)* Validation accu-
tion                                                    racy

*Figure 8.* Offline post-check simulation outcomes

Section 5.

### 4.5. Evidence 5: Anchor Scores Provide a Borderline Ranking Signal

*Evidence 5: In the borderline region (scores 4–6), papers prioritized by anchor scores exhibit higher future citations than those prioritized by reviewer scores, using citations only as a coarse ex-post proxy.*

**Setup.** We focus on the **borderline zone** (scores 4–6), where decisions are most contentious and calibration matters most. Using ICLR 2023 ($N = 3{,}507$) and 2024 ($N = 7{,}150$), we partition papers into the same score bands under two regimes: (i) bands defined by reviewer-average scores and (ii) bands defined by anchor scores. Within each borderline sub-band, we compare mean citation counts between the two regimes. We treat citations as a coarse, ex-post signal of impact rather than a ground-truth quality label, and report them only as supportive evidence about ranking behavior. Since such ex-post evidence may raise hindsight or contamination concerns, we evaluate the same ranking logic under older-model and post-cutoff settings (Appendices C.4 and F.3).

**Findings.** Within the borderline sub-bands, anchor-selected papers consistently achieve higher average citations than reviewer-selected papers (Figure 9). This suggests that when outcomes hinge on fine-grained score differences, the rationale-grounded anchor score can better preserve quality signals implicit in reviews that are not reliably reflected in raw numeric ratings. While citations are imperfect and may reflect field size or exposure effects, the directionality is aligned with our position: calibration is most valuable near the boundary, and in that regime anchor scoring appears to produce a more informative ordering than the unconstrained human score interface.

### 4.6. Boundary-Focused Validity Checks

*Boundary checks: Quasi-experimental proxy analyses reduce the concern that residuals merely proxy paper quality, but they do not prove causal improvement in review correctness.*

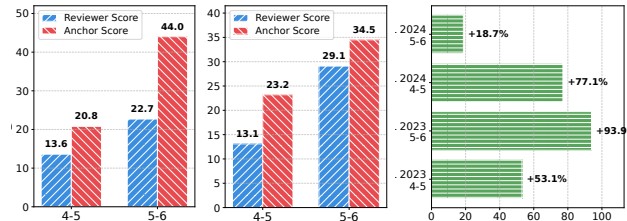

*(a)* ICLR 2023 cita-*(b)* ICLR 2024 cita-*(c)* Citation improvement
tions                  tions

*Figure 9.* Borderline (4–6) citation outcomes by scoring regime

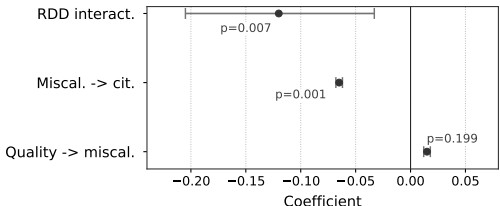

*Figure 10.* Boundary-focused validity checks

**Setup.** We use boundary-focused proxy analyses in Appendix F.4: an RDD moderator check around the accept/reject boundary and a mediation analysis separating paper-quality proxies from miscalibration. These checks are best interpreted as identification-oriented robustness tests, not causal validation of the calibration mechanism.

**Findings.** The boundary checks are directionally consistent (Figure 10): in the ICLR 2024 RDD, larger residual magnitude reduces the citation gain from acceptance near the boundary (treated $\times |residual| = -0.119$, $p = 0.007$). Mediation analysis finds that paper quality does not significantly predict miscalibration ($p = 0.199$), while miscalibration remains negatively associated with citation outcomes after controlling for quality ($p = 0.001$). Together with the leakage, prompt, cross-model, and AI-era stress tests above, these results support using residuals as triggers for human clarification, not as ground-truth labels of scientific merit.

### 4.7. Summary of Evidence

Taken together, the five pieces of evidence support our central claim that rationale–score misalignment is measurable and actionable at conference scale. Evidence 1 establishes that anchor scoring is stable under fixed inputs and responsive to rationale perturbations; Evidence 2 shows growing residual dispersion; Evidence 3 finds that extreme disagreements concentrate on interpretable dimensions such as evidence strength and novelty; Evidence 4 shows that post-checks can plausibly reconcile large mismatches; and Evidence 5 links anchor-based borderline ranking with higher downstream citation impact. The boundary-focused checks further reduce the concern that residuals merely proxy paper

quality, but they should be read as robustness checks rather than causal proof. The leakage-aware checks do not eliminate all validity concerns, but they address the strongest alternative explanation that the empirical signal is primarily a memorization artifact. Overall, these results motivate treating scores as an auditable interface: residuals provide a concrete signal for targeted clarification without replacing human scientific judgment.

## 5. Discussions

**Human-in-the-loop validation.** Our post-check evidence (Section 4.4) is based on offline simulation, so the next step is a low-risk **shadow audit** or **opt-in post-check** pilot measuring time cost, revision-versus-justification rates, justification specificity, area-chair usefulness, and compliance bias.

**Leakage, privacy, and reproducibility.** Data leakage is a central validity risk for historical OpenReview analysis, so we treat ICLR 2023–2025 as one evidence source and add memorization screens, older-model reruns, post-cutoff datasets, and gpt-5.4-mini variance controls (Appendices A.2 and C.4). For deployment, venues should avoid third-party APIs by default, pin the model, prompt, decoding settings, and rubric on conference-managed infrastructure, and log artifacts for replay.

**Gaming, scope control, and beyond scoring.** Residual-triggered escalation can be optimized against by vague rationales or author-side gaming, so calibration should remain an audit-and-escalation mechanism rather than a primary ranking rule. The model should output auditable signals, not verdicts; the same principle can extend to verification-oriented flags such as claim–evidence mismatches or missing cited baselines.

## 6. Alternative Views

**Some inconsistency is desirable; calibration risks over-standardization.** Reviewers may legitimately weight novelty, evidence strength, clarity, and long-term potential differently; calibration could therefore suppress pluralism or create false objectivity. Our response is that the layer audits rationale–score coherence, not merit itself: inconsistent scores trigger justification or revision, while our diversity-preservation check on ICLR 2026 keeps 74.7% of score diversity and leaves 96.4% of semantically different reviewer pairs with distinct scores (Appendix C.5). The remaining differences are precisely the cases that should stay visible to area chairs as substantive disagreement rather than be collapsed into a single model judgment.

**LLMs may be biased; a calibration layer may import**

**new failure modes.** LLMs can reflect training-data biases, respond to prompt details, and introduce new attack surfaces. We therefore bound the model's role: it never makes accept/reject decisions, never overrides scores, and only produces an anchor and residual from paper content and structured review fields under a pinned, replayable protocol. Prompt rephrasing is stable under semantically equivalent variants ($ICC(2, 1) = 0.9493$), but explicitly strict or lenient prompts shift outputs, making prompt pinning a governance requirement. This limitation is not incidental; it is why the proposed system is an auditing layer rather than an autonomous reviewer.

**LLM-assisted authors and reviewers may make calibration redundant or gameable.** Authors may optimize manuscripts for LLM-facing signals while reviewers may use LLMs to draft rationales or scores. Our stress tests suggest that this does not remove the need for venue control: writing optimization leaves anchors stable when human pros/cons are fixed (Pearson $r = 0.9016$; 90.4% change by at most one point), while mixed human/LLM reviewer simulations still show substantial bias span from heterogeneous tools, prompts, and models.

**Why not adopt an end-to-end AI review system?** If an LLM can standardize review-text-to-score mapping, one might ask why not automate review entirely. Full automation is premature: end-to-end LLM judging may favor training-distribution patterns, lacks accountability, and invites optimization against model preferences. We advocate the narrower audit-and-escalation role: preserve human judgment while making score comparability explicit.

## 7. Call to Action

Turning residual-based calibration into a real conference capability requires coordinated action across roles rather than an one-off tool integration. The near-term goal is modest: make the scoring step more comparable and auditable, while keeping scientific judgment and accept/reject authority with reviewers and area chairs.

**Conference organizers.** Start with a shadow audit: compute anchors and residuals for first-round reviews, but do not expose them to reviewers or affect decisions. This phase should measure trigger rates, concentration by area and score band, operational cost, and agreement with area-chair perceptions of unclear reviews. The pilot should pre-register the leakage and privacy protocol: model version, processed data fields, third-party-service status, and retained replay logs.

**Area chairs.** Treat residuals as triage, not decisions. When a review has a large mismatch, the response is reconciliation: the reviewer either revises the score to match the

rationale or keeps the score with a short, paper-grounded justification. This makes the numeric interface easier to interpret near the boundary while leaving substantive disagreement to normal AC deliberation.

**Platform maintainers and infrastructure owners.** Record the artifacts needed for audit and replay: paper version, rationale fields, extracted rationale items, anchor score, item-level contributions, residual, and trigger outcome. Because manuscripts and reviews are confidential before publication, deployments should favor conference-managed or self-hosted infrastructure, strict access control, and retention policies that separate aggregate audit statistics from identifiable review content.

**Steering committees and policy groups.** Make the boundary explicit: the model supports auditing and escalation, does not generate accept/reject recommendations, and never automatically overrides human judgments. Any protocol adaptation should occur only between cycles with documented version changes. Governance reports should disclose calibration reliability, leakage-control checks, trigger rates, reviewer response patterns, and known failure modes.

## 8. Related Works

**Inconsistency and rater effects in peer review.** Peer review is noisy and sensitive to reviewer assignment: NeurIPS consistency experiments attribute substantial accept/reject variance to reviewer disagreement (Cortes & Lawrence, 2021; Beygelzimer et al., 2023), while rater-effect studies show that individual scoring scales are arbitrary or miscalibrated (Sculley et al., 2018; Stelmakh et al., 2021; Ammar & Shah, 2012; Mitliagkas et al., 2011). Recent evidence further suggests that commensuration bias—how reviewers translate multi-dimensional criteria into one score—is a major source of divergence (Goldberg et al., 2025). Our work targets this specific failure mode by auditing rationale–score coherence rather than eliminating disagreement about merit.

**AI-assisted peer review.** Prior work predicts decisions or ratings from historical reviews (Kang et al., 2018; Fernandes & Vaz-de Melo, 2022; Bharti et al., 2021) or explores LLMs as reviewers and review simulators (Tan et al., 2024; Li et al., 2025; Gao et al., 2025; Chang et al., 2025; Idahl & Ahmadi, 2025; Jin et al., 2024). These systems can aid triage but risk inheriting rater effects, prompt injection (Keuper, 2025), and systematic scoring shifts (Latona et al., 2024); we instead map human rationales into auditable anchors for calibration. This distinction matters operationally: the calibration layer does not learn from historical accept/reject labels and does not generate new scientific critiques. The anchor is computed from reviewer-authored strengths and

weaknesses, so disagreement about scientific merit remains in the review text and AC discussion. The model only checks whether the numeric score attached to that text follows a common rubric.

**LLM as a judge.** LLM-based evaluators are increasingly used when automatic metrics are insufficient: MT-Bench documents judge biases and mitigations (Zheng et al., 2023), G-Eval shows benefits from rubric-grounded prompting (Liu et al., 2023), and surveys emphasize reliability risks (Gu et al., 2026). These lessons motivate our design: explicit rubrics, logged intermediate artifacts, pinned prompts, and residuals as workflow signals rather than autonomous decisions. Thus the relevant reliability question is narrower than whether LLMs can review papers. It is whether a fixed judge can reproduce a documented rationale-to-score mapping under stable prompts, which is why our validation focuses on leakage checks, prompt sensitivity, and cross-model reuse.

**Review-workflow interventions.** Recent discussions of AI conference reviewing emphasize that scale pressure, reviewer heterogeneity, and uneven AC bandwidth are process-level problems rather than only modeling problems (Kargaran et al., 2025; Aczel et al., 2025; Kim et al., 2025). This motivates interventions that expose where human attention is most needed instead of adding another opaque score. Our residual signal fits this line of work: it is designed to prioritize review-level anomalies for follow-up, while leaving the final interpretation to reviewers and area chairs. Because it can run before AC deliberation, it also permits shadow-mode measurement of trigger rates and reviewer response patterns before any policy change.

## 9. Conclusion

Using OpenReview data from ICLR 2023–2025, we identify rationale–score calibration as a concrete bottleneck for peer review at scale. We find measurable scale drift: similar rationales can yield non-comparable scores, and rationale–score misalignment appears to worsen as submission volumes grow. Leakage-aware reruns, post-cutoff datasets, and cross-model checks reduce the likelihood that the main signal is a memorization artifact, although they do not replace a prospective human pilot. We argue that AI conference workflows should add an LLM-driven calibration layer that standardizes the rationale-to-score mapping under a fixed rubric and uses residuals as a selective audit signal, while keeping scientific judgment with humans. The practical path is to start with shadow auditing, fixed protocols, privacy-preserving infrastructure, and opt-in post-checks before any stronger workflow integration. By making the scoring step more auditable and internally consistent, such a layer can expose review-level mismatches that area chairs already need to resolve, without turning the model into a decision-maker.

## Acknowledgements

This work is supported by National Natural Science Foundation of China (Grant No. 62502174), Fundamental and Interdisciplinary Disciplines Breakthrough Plan of the Ministry of Education of China (No. JYB2025XDXM118), National Natural Science Foundation of China (Grant No. U25B2023 and Grant No. 62322205), and Hubei Provincial Natural Science Foundation of China (No. 2026AFA002).

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

# A. Additional Method Details

## A.1. Data and Review Scope

We study all ICLR 2023–2025 submissions available on OpenReview, including both accepted and rejected papers. We restrict our analysis to **first-round reviews only**, i.e., the initial reviews prior to rebuttal and any subsequent score edits. For each review, we use the `strengths`, `weaknesses`, and `overall rating` fields, where the overall rating is on a 1–10 scale. We do not use rebuttal-stage updates, discussion threads, meta-reviews, or area-chair comments. Paper-level dispersion statistics (Appendix A.3.3) are computed on papers with at least two first-round reviews.

## A.2. Leakage-Aware Filtering

To reduce the risk of evaluation-set leakage due to LLM memorization, we apply conservative filtering procedures.

**ICLR 2025 time-cutoff protocol.**    For ICLR 2025, we follow a time-cutoff protocol commonly adopted in large-scale empirical studies of LLM feedback on research papers: we use models whose release dates precede the public release of ICLR 2025 reviews (Liang et al., 2024b). This cutoff reduces the likelihood that the model has directly observed ICLR 2025 review text during training.

**Memorization screens for ICLR 2023–2024.**    For ICLR 2023–2024, where a strict time cutoff is infeasible, we apply two complementary memorization screens inspired by prior work on quantifying memorization and detecting pretraining-data contamination (Carlini et al., 2023; Shi et al., 2024).

**Screen 1 (summary continuation overlap).** We provide the LLM with the first half of a paper summary and ask it to complete the second half. If the completion exhibits high overlap with the true continuation, we treat the paper as likely memorized and remove it from analysis.

**Screen 2 (title-based factual recall).** We provide the paper title and ask three factual questions: the first author, the affiliation, and the venue. If the LLM answers at least two of these questions correctly, we conservatively filter the paper as likely memorized.

**Additional leakage controls.**    Beyond these filtering steps, we further stress-test the main findings under lower-contamination settings: older-model reruns for ICLR 2023–2024, post-cutoff and newer conference datasets, a gpt-5.4-mini variance control, and a separate citation-analysis leakage control (Appendices B.4, C.4, and F.3). These checks keep the scoring pipeline fixed while varying the model or dataset where appropriate, and assess whether the observed residual-dispersion and downstream patterns persist when memorization-based explanations are less plausible.

**Conservative stance.**    We treat these procedures as safeguards that prioritize avoiding leakage, even at the cost of filtering out some non-memorized cases.

## A.3. Pipeline Method Details

### A.3.1. RATIONALE EXTRACTION

For each paper $p$ and reviewer $j$, we convert the free-form `strengths` and `weaknesses` fields into itemized rationale statements using an LLM. Let $\mathcal{S}_{p,j}$ and $\mathcal{W}_{p,j}$ denote the extracted strength and weakness items, respectively. This step standardizes heterogeneous writing styles into comparable atomic claims that can be mapped to a rubric-consistent score.

**Output form.**    The extractor outputs two lists of short statements: a list of strengths $\mathcal{S}_{p,j}$ and a list of weaknesses $\mathcal{W}_{p,j}$. Each item is intended to be a single, self-contained rationale (e.g., one claim about novelty, evidence strength, clarity, or missing experiments), so that downstream scoring can attribute score contributions at the item level.

### A.3.2. RUBRIC-GROUNDED ANCHOR SCORING

Given the paper content and the extracted rationale items, we compute an anchor score $s_{p,j}^A$ that represents a rubric-consistent translation from "what the reviewer wrote" to "what score that implies" under a fixed rubric and prompting protocol. The LLM assigns each strength item $s \in \mathcal{S}_{p,j}$ a bonus $\Delta^+(s) \in [0,5]$ and each weakness item $w \in \mathcal{W}_{p,j}$ a penalty

$\Delta^-(w) \in [-5, 0]$, where the magnitude represents the estimated contribution of that rationale to the overall rating. We then aggregate these contributions with a midpoint base score and clip to the ICLR 1–10 scale:

$$s_{p,j}^A = \text{clip}\left(s_0 + \sum_{s \in \mathcal{S}_{p,j}} \Delta^+(s) + \sum_{w \in \mathcal{W}_{p,j}} \Delta^-(w), \ 1, \ 10\right) \tag{1}$$

where we use $s_0 = 5$. This yields an auditable, rationale-grounded reference score under a fixed rubric and prompting protocol.

**Interpretation.** We emphasize that the anchor score is used as a consistent reference mapping for auditing coherence between the reported score and the written rationale. It is not treated as a ground-truth label of paper quality.

A.3.3. RESIDUALS AND PAPER-LEVEL DISPERSION METRICS

Let $s_{p,j}^H \in \{1, \ldots, 10\}$ be the reviewer's reported overall rating. We define the per-review residual:

$$r_{p,j} = s_{p,j}^H - s_{p,j}^A \tag{2}$$

Residuals quantify rationale–score misalignment at the review level.

**Paper-level aggregation.** To measure how differently multiple reviewers translate their rationales into scores for the same paper, we aggregate residuals at the paper level. For each paper $p$ with reviewer set $\mathcal{R}(p)$, we compute:

$$\sigma_p = \text{Std}\left(\{r_{p,j}\}_{j \in \mathcal{R}(p)}\right) \tag{3}$$

$$\text{Range}_p = \max_{j \in \mathcal{R}(p)} r_{p,j} - \min_{j \in \mathcal{R}(p)} r_{p,j} \tag{4}$$

Here, $\sigma_p$ captures dispersion in rationale–score residuals across reviewers, while $\text{Range}_p$ provides an interpretable maximum-disagreement measure. Conference-level trends are summarized by aggregating these paper-level statistics within each year.

## B. Experiment 1: Test-Retest Reliability Details

This appendix details the experimental procedure and specific prompts used for the test-retest reliability analysis of the anchor scoring pipeline.

### B.1. Experimental Procedure

The reliability analysis was conducted following a systematic protocol to quantify stochastic noise in our scoring pipeline. To evaluate the consistency of the anchor mapping, we performed a test-retest reliability analysis on a subset of papers from the ICLR 2023, 2024, and 2025 conferences. We sampled approximately 20% for ICLR 2023, ICLR 2024, and ICLR 2025 ($N = 701, 716, 2304$ respectively). For each paper, we obtained $K$ independent ratings ($K = 3$ for 2023/2024/2025), where one rating is the original score from the dataset and the remaining $K - 1$ are generated by our pipeline. This setup allows us to measure both the stochasticity of the LLM generation and the reconstruction fidelity of the pipeline. We report the mean variance across parallel runs to quantify the instability:

$$\text{Var} = \frac{1}{R} \sum_{i=1}^{R} \text{Var}(s_{i,1}, \ldots, s_{i,K}) \tag{5}$$

where $R$ is the total number of reviews analyzed. As shown in Table 1, the variance remains low ($< 0.13$) for 2023 and 2024, but increases for 2025, suggesting potential sensitivity to the larger scale or changes in formatting. All executions were performed using the same large language model (GPT-4o-mini) under identical parameters (Temperature = 0.1) to ensure that observed variance is purely reflective of the model's internal stochasticity in quantization and reasoning. We specifically focus on the re-quantization step, where qualitative features (pros and cons) are mapped to numerical weights. The variance and pairwise correlation of the final scores across the parallel runs were calculated to assess the stability of this mapping.

## B.2. Summary Results

Table 1 presents the quantitative results of the test-retest reliability analysis across ICLR 2023, 2024, and 2025. The table aggregates the total number of papers sampled, the volume of reviews processed ($R$), and the number of independent scoring iterations ($K$). The reported mean variance serves as a metric for the stability of the anchor scoring pipeline across different years.

*Table 1.* Test-retest reliability summary

| Conference | Papers | Reviews ($R$) | Repeats ($K$) | Mean Variance |
|---|---|---|---|---|
| ICLR 2023 | 701 | 2604 | 3 | 0.1020 |
| ICLR 2024 | 716 | 2709 | 3 | 0.1054 |
| ICLR 2025 | 2304 | 8827 | 3 | 0.5241 |

## B.3. Controlled Rationale Intervention for Evidence 1

**Controlled rationale intervention.**    To test whether residuals respond to rationale–score consistency rather than paper-level disagreement alone, we hold paper content and the original reviewer score fixed while modifying only the rationale text for 200 papers each from ICLR 2023, 2024, and 2025. More supportive rationales move the anchor closer to the assigned score; less supportive and generic boilerplate rationales increase misalignment.

*Table 2.* Controlled rationale intervention with paper content and original score fixed

| Rationale Version | Mean Absolute Residual |
|---|---|
| More supportive of the assigned score | 0.641 |
| Less supportive of the assigned score | 2.573 |
| Generic boilerplate only | 1.863 |
| Original rationale | 1.333 |

## B.4. ICLR 2025 Variance Surge Control

**ICLR 2025 variance surge control.**    One concern is that Figure 4 could reflect a shift from potentially leaked ICLR 2023–2024 data to non-leaked ICLR 2025 data. To isolate this factor, we repeat the same variance analysis using gpt-5.4-mini, whose pretraining explicitly includes ICLR 2023–2025 data. This control uses the same test–retest inputs, prompts, decoding settings, and variance computation as Evidence 1, with gpt-5.4-mini substituted for the original anchor model. Even under this setting, the ICLR 2025 variance remains much larger than 2023 and 2024. This suggests that the variance increase is not explained by the leaked/non-leaked transition alone, and is more consistent with increased submission scale, quality heterogeneity, and scoring inconsistency.

*Table 3.* Variance control using gpt-5.4-mini across ICLR 2023–2025

| Year | Mean Variance | Median Variance |
|---|---|---|
| 2023 | 0.0830 | 0.0300 |
| 2024 | 0.0906 | 0.0400 |
| 2025 | 0.3496 | 0.2755 |

# C. Experiment 2: Review Bias Analysis Framework

This appendix provides a detailed description of the experimental pipeline and the specific LLM prompts used to quantify review bias.

## C.1. Experimental Pipeline

The bias analysis follows a four-stage process to systematically evaluate the discrepancy between generated and actual scores:

1. **Data Collection:** Submissions and reviews are ingested from OpenReview.

2. **Feature Extraction:** For each review, an LLM extracts granular strengths (Pros) and weaknesses (Cons).

3. **Weight Quantization:** The LLM assigns numerical weights to each extracted feature based on the paper content and abstract.

4. **Bias Calculation:** The expected score ($S_{expected} = S_{base} + \sum W_{pros} + \sum W_{cons}$) is compared against the actual score ($S_{actual}$).

## C.2. LLM Prompts (Raw Templates)

### C.2.1. FEATURE EXTRACTION PROMPT (EXTRACT_PROS_CONS_BATCH)

The following is the English translation of the prompt template used for the batch feature extraction step in Stage 2.

---

**SYSTEM_PROMPT = ""”"**
*You are a senior academic review expert. Please carefully read all the review comments for the following paper and extract Pros and Cons for each reviewer.*

**Paper Title:** {title}
**Paper Abstract:** {abstract}

Below are the review comments from {num_reviewers} reviewers:
{all_reviews_text}

**Requirements:**
1. Extract Pros and Cons for each reviewer respectively, maintaining the correspondence with reviewer IDs
2. Extract each pro and con and summarize them briefly (1-2 sentences) [cite: 18]
3. Assign an appropriate category to each pro/con from the following list: {categories} [cite: 18]
4. Pros and cons should be specific and quantifiable
5. Return the results in JSON format

**Example return format (assuming 2 reviewers):**

```
{
  "reviewers": [
    {
      "reviewer_id": "Reviewer_A",
      "pros": [
        {
          "description": "Proposed an innovative attention mechanism",
          "category": "Novelty/Originality"
        }
      ],
      "cons": [
        {
          "description": "Lack of comparison with latest SOTA methods",
          "category": "Experimental Rigor"
        }
      ]
    },
    {
      "reviewer_id": "Reviewer_B",
      "pros": [
        {
          "description": "Rigorous experimental design",
          "category": "Experimental Rigor"
```

---

```
            }
        ],
        "cons": [
            {
                "description": "Writing contains grammatical errors",
                "category": "Writing Quality"
            }
        ]
    }
  ]
}
```

Please strictly follow the JSON format above, ensuring each reviewer's reviewer_id matches the input exactly.
"""

*Figure 11.* Prompt used in our analysis for feature extraction

### C.2.2. WEIGHT QUANTIZATION PROMPT (QUANTIFY_WEIGHTS)

The following prompt template is used for the weight quantization step in Stage 3.

**SYSTEM_PROMPT = """**
*You are an objective and fair academic review expert. Now you need to assign quantitative weights to the pros and cons extracted from the review, combined with the full content of the paper.*

**Paper Title:** {title}
**Paper Abstract:** {abstract}

**Full Paper Content (for reference):**
{paper_content}

**Identified Pros (total {pros_count}):**
{pros_text}

**Identified Cons (total {cons_count}):**
{cons_text}

**Scoring Criteria:**
- Score range: {min_score} to {max_score}
- Base Score: {base_score} (representing an average quality paper)
- Pro weight (positive): represents how much the pro should increase the total score
- Con weight (negative): represents how much the con should decrease the total score

**Weight Allocation Principles:**
1. Core elements such as novelty and technical correctness should be given higher weights ($\pm0.5$ to $\pm2.0$) [cite: 18]
2. Secondary factors such as writing quality and formatting issues should be given lower weights ($\pm0.1$ to $\pm0.5$) [cite: 18]
3. Fatal flaws (e.g., theoretical errors) can be given extremely high negative weights (-2.0 or lower) [cite: 18]
4. Significant innovations can be given extremely high positive weights (+2.0 or higher) [cite: 18]
5. The total sum of weights should be reasonable, so that the expected score falls within the range of {min_score} to {max_score} [cite: 18]

**\*\*Important Requirements\*\*:** [cite: 24, 25]
- You must assign a weight to [each] pro and con listed above, without omitting any
- The pros_weights array must contain exactly {pros_count} elements
- The cons_weights array must contain exactly {cons_count} elements
- Even if some entries look similar or redundant, you must assign weights to each one individually
- Output the weights sequentially in the order they were input

**Please return in JSON format:** [cite: 27]

```
{
```

```
    "pros_weights": [
      {
        "description": "Description of Pro 1",
        "category": "Category",
        "weight": 1.5,
        "reasoning": "Reasoning for weight allocation"
      }
    ],
    "cons_weights": [
      {
        "description": "Description of Con 1",
        "category": "Category",
        "weight": -1.0,
        "reasoning": "Reasoning for weight allocation"
      }
    ],
    "expected_score_breakdown": {
      "base_score": {base_score},
      "total_pros_weight": 0.0,
      "total_cons_weight": 0.0,
      "expected_score": 0.0
    }
  }
"""
```

*Figure 12.* Prompt used in our analysis for weight quantization

## C.3. Aggregated Results

Table 4 presents the comprehensive statistics regarding the alignment between anchor-generated scores and actual reviewer scores across the three conferences. The data highlights trends in Mean Bias, *Mean Absolute Error* (MAE), and the frequency of extreme scoring divergences ($|r| > 1.5$), providing a quantitative basis for the observations discussed in Section 4.2.

*Table 4.* Comparative bias statistics for ICLR 2023–2025

| Metric | ICLR 2023 | ICLR 2024 | ICLR 2025 |
|---|---|---|---|
| Papers (Total) | 3,507 | 7,150 | 11,520 |
| Mean Bias | +0.260 | -0.232 | -1.188 |
| MAE (Mean Abs. Residual) | 1.357 | 1.362 | 1.714 |
| Bias Span (Mean) | 3.007 | 3.064 | 3.211 |
| Bias Span (Max) | 9.5 | 10.5 | 11.5 |
| Extreme Bias Ratio ($|r| > 1.5$) | 32.9% | 32.9% | 44.8% |
| $r$ (Actual vs Exp.) | 0.483 | 0.478 | 0.437 |

## C.4. Additional Leakage and Contamination Controls

Because LLM-based scoring may be affected by pretraining contamination, we run additional controls for the residual-dispersion analysis. These controls test whether the Evidence 2 dispersion pattern survives under lower-contamination settings rather than certifying that no individual training example is observed by a model. Unless a control explicitly states otherwise, it reuses the same rationale-extraction prompt, anchor-scoring prompt, decoding settings, parsing rules, residual definition, and paper-level aggregation metrics as the main pipeline described in Appendices A.3.1–A.3.3. Thus, these controls vary only the model or dataset named in each paragraph, not the scoring rubric or post-processing procedure.

**Older-model reruns on ICLR 2023–2024.** We rerun the key residual-dispersion analysis using gpt-3.5-turbo-1106, whose release timing precedes the public release of the target papers and reviews. The rerun keeps the main pipeline fixed and

replaces only the anchor-scoring model with gpt-3.5-turbo-1106. The main dispersion pattern remains substantial in both years.

*Table 5.* Older-model reruns for leakage control on ICLR 2023–2024

| Dataset | Bias Span Mean | Mean Residual Std. Dev. |
|---------|----------------|-------------------------|
| ICLR 2023 | 2.665 | 1.726 |
| ICLR 2024 | 2.691 | 1.715 |

**Post-cutoff datasets.** We also evaluate data released after the anchor model's knowledge cutoff, including ICLR 2026, ICML 2025, and NeurIPS 2025. These datasets provide a lower-contamination testbed for the same residual-dispersion claim. For this control, the extraction, anchor-scoring, and residual-aggregation procedures are unchanged; only the conference dataset is replaced.

*Table 6.* Residual dispersion on post-cutoff or newer datasets

| Dataset | Bias Span Mean | Mean Residual Std. Dev. |
|---------|----------------|-------------------------|
| ICLR 2026 | 3.601 | 1.643 |
| ICML 2025 (partial) | 3.950 | 1.826 |
| NeurIPS 2025 (partial) | 2.980 | 1.337 |

### C.5. Prompt Sensitivity and AI-Era Stress Tests

**Prompt rephrasing.** We evaluate four semantically equivalent prompt variants for the rationale-to-score conversion step: the original Chinese academic prompt, a concise instruction prompt, an area-chair role-play prompt, and a structured step-by-step prompt. All variants encode the same rubric and scoring ranges, differing only in wording and presentation style. The resulting scores achieve $\text{ICC}(2,1) = 0.9493$, indicating high stability under ordinary prompt rephrasing. However, prompts that explicitly instruct the model to score leniently or strictly shift the outputs, so deployment should pin the exact prompt template for an entire review cycle.

**Prompt-rephrasing experimental protocol.** The prompt-sensitivity experiment is run on a random 20% sample of ICLR 2026 submissions with a fixed random seed. All prompt variants share the same extracted pros/cons and the same anonymized reviewer-item mapping; only the Stage-3 rationale-to-score prompt changes. For each paper, pros and cons are shuffled across reviewers before weight assignment and then mapped back to the original reviewers for expected-score reconstruction. The model is GPT-4o-mini with temperature 0.1, a 1–10 score range, and base score 5.0. The paper text supplied to the weight prompt is capped at 15,000 characters. For each variant, we compute expected reviewer scores, pairwise Pearson/Spearman correlations, pairwise MAE, cross-variant score dispersion, and $\text{ICC}(2,1)$.

**Prompt variants used in the rephrasing test.** Variant A is the original weight-quantization template reported in Appendix C. Variants B–D keep the same input fields, scoring range, base score, completeness constraints, and JSON schema, but alter the instruction style as follows.

---

**Variant B: concise instruction style.**

```
Assign quantitative weights to the pros and cons in the following reviews.

Paper: {title}
Abstract: {abstract}
Full paper: {paper_content}

Pros ({pros_count} items):
{pros_text}

Cons ({cons_count} items):
```

---

```
{cons_text}

Rules:
- Base score: {base_score}; range: [{min_score}, {max_score}]
- Pros receive positive weights; cons receive negative weights.
- Core issues such as novelty and technical correctness: +/-0.5 to +/-2.0.
- Secondary issues such as writing and formatting: +/-0.1 to +/-0.5.
- Fatal flaws may be -2.0 or lower; major breakthroughs may be +2.0 or higher.
- Return exactly {pros_count} pro weights and {cons_count} con weights.
- Do not omit any item.

Return JSON with pros_weights, cons_weights, and expected_score_breakdown.
```

*Figure 13.* Concise instruction variant used in the prompt-rephrasing sensitivity test

```
Variant C: area-chair role-play style.

You are an Area Chair at a top AI conference and are calibrating reviewer scores.
Based on the paper content, independently evaluate the impact of each review item
and assign it a quantitative weight.

Paper title: {title}
Paper abstract: {abstract}
Full paper for reference: {paper_content}

Reviewer pros ({pros_count} items):
{pros_text}

Reviewer cons ({cons_count} items):
{cons_text}

As Area Chair:
- Start from {base_score}, representing an average paper.
- Use the score range {min_score}-{max_score}.
- Assign positive weights to pros and negative weights to cons.
- Key contributions such as methodological novelty or theoretical breakthroughs
  should receive +0.5 to +2.0 or higher.
- Serious problems such as incorrect proofs or invalid experiments should receive
  -0.5 to -2.0 or lower.
- Surface-level issues such as writing polish or formatting should receive
  weights around +/-0.1 to +/-0.5.
- The reconstructed expected score should remain within the valid range.

Return exactly {pros_count} pro weights and {cons_count} con weights.
Keep the original input order.
Return only the required JSON object.
```

*Figure 14.* Area-chair role-play variant used in the prompt-rephrasing sensitivity test

```
Variant D: structured step-by-step instruction style.

You are an academic review expert. Assign quantitative weights to each pro and con
by following these steps.

Paper title: {title}
Paper abstract: {abstract}
Full paper: {paper_content}

Step 1: Review the pros ({pros_count} items):
```

```
{pros_text}

Step 2: Review the cons ({cons_count} items):
{cons_text}

Step 3: Estimate the actual impact of each item on paper quality.
The base score is {base_score}, representing an average paper, and the valid score
range is {min_score} to {max_score}.

Step 4: Use these weight standards:
- Core contributions or core flaws: +/-0.5 to +/-2.0.
- Secondary issues such as writing quality, formatting, and readability:
  +/-0.1 to +/-0.5.
- Fatal flaws may be -2.0 or lower.
- Major innovations may be +2.0 or higher.

Step 5: Give a brief reason for each assigned weight.
Step 6: Verify that the base score plus all weights falls within the valid range.

Return exactly {pros_count} pro weights and {cons_count} con weights.
Keep the original input order.
Return only JSON.
```

*Figure 15.* Structured step-by-step variant used in the prompt-rephrasing sensitivity test

**Author-side LLM optimization.** To test whether authors can easily optimize papers for the calibration layer, we sample 200 ICLR 2026 papers and use GPT-4o-mini to revise writing style and presentation without changing technical content. We then rerun the calibration pipeline using the revised paper and the original human reviewer pros/cons. The original and optimized anchor scores remain highly correlated (Pearson $r = 0.9016$), and 90.4% of anchor scores change by at most one point on the 1–10 scale. This indicates that the anchor is primarily driven by the reviewer's stated rationale rather than paper style alone.

**Author-side optimization experimental protocol.** This experiment simulates an author who knows that the review workflow contains an AI calibration layer. For each of 200 randomly sampled ICLR 2026 submissions, we first extract up to 15,000 characters of paper text. GPT-4o-mini is then called twice: first as an AI academic-review consultant that proposes writing changes likely to improve AI-review presentation, and second as an author that rewrites the paper's core content according to those suggestions. The rewritten text is used only as the paper-content input to the same weight-quantization step; the original human reviewer pros/cons remain fixed. This isolates whether surface-level author-side optimization changes the anchor score when the reviewer rationale is unchanged. We compare the original and optimized anchor scores using Pearson correlation, ICC, MAE, paired $t$-test, and the distribution of score changes.

**Author-side optimization prompts.** The following are English translations of the two author-side prompts used before rerunning weight quantization.

```
Prompt 1: simulated AI-review consultant.

You are an experienced AI academic-review consultant. An author plans to submit
this paper to a review workflow that includes AI calibration scoring. Please
simulate reviewing this paper and focus on concrete writing-strategy suggestions
that could help the paper perform better in an AI-review process.

Paper title: {title}

Paper content:
{paper_content}

Please provide:
1. The paper's main strengths and weaknesses as a simulated review.
```

```
2. Five to ten specific writing-optimization suggestions, focusing on how to make
   the AI system better recognize the paper's contribution and value.
3. Specific paragraphs or parts that should be revised and the revision direction.

Return the suggestions as clearly structured text.
```

*Figure 16.* Simulated AI-review consultant prompt used for author-side optimization

```
Prompt 2: simulated author revision.

You are an academic-paper author. Based on the following review comments and
optimization suggestions, revise the paper in a targeted way. Generate the
revised core paper content, including the abstract, introduction, method overview,
experiment summary, and conclusion.

Original paper title: {title}

Original paper content:
{paper_content}

Review comments and optimization suggestions:
{review_suggestions}

Please output the revised core paper content. Requirements:
1. Keep a formal academic writing style.
2. Incorporate the review suggestions into the revision.
3. Strengthen the descriptions of contributions and experimental design.
4. The output should be complete paper text, not a revision explanation.
```

*Figure 17.* Simulated author-revision prompt used for author-side optimization

**Mixed human/AI reviewer behavior.** We simulate four reviewer types on 1,000 randomly sampled ICLR 2026 papers: fully human reviews (H), human pros/cons with AI-assigned scores (T1), AI-rewritten pros/cons with human scores (T2), and fully AI-generated reviews (T3). The AI reviewer role is randomly assigned among GPT-4o-mini, Claude-3.5-Sonnet, and Gemini-2.5-Flash, while calibration is performed uniformly with GPT-4o-mini. Bias span remains substantial across all mixtures, including the setting where all reviewers use LLMs.

*Table 7.* Bias span under mixed human/AI reviewer behavior

| Scenario | H | T1 | T2 | T3 | Bias Span Mean |
|---|---|---|---|---|---|
| A (baseline) | 100% | 0% | 0% | 0% | 3.463 |
| B (light LLM) | 60% | 15% | 15% | 10% | 3.622 |
| C (moderate LLM) | 30% | 20% | 30% | 20% | 3.539 |
| D (heavy LLM) | 0% | 25% | 25% | 50% | 3.001 |

**Diversity preservation.** To test whether calibration suppresses genuine opinion diversity, we compute pairwise semantic similarity of reviewer pros/cons using SBERT and identify 11,164 reviewer pairs with semantically different viewpoints and different scores. Calibration reduces mean score difference from 2.733 to 2.042, preserving 74.7% of the original score diversity. After calibration, 96.4% of these pairs still retain distinct scores. The estimated correctable potential bias is 1.67 points per reviewer, suggesting that the method removes part of the scale noise while retaining most substantive disagreement.

## D. Experiment 3: Qualitative Analysis of Extreme Divergence

This appendix presents the detailed methodology and results of the qualitative analysis performed on papers exhibiting extreme divergence in reviewer scores (high bias span).

### D.1. Experimental Procedure

To investigate the sources of disagreement in the review process, we focused on the subset of papers with the highest discrepancies between reviewer scores and our model's expected scores. The analysis proceeded in three stages:

1. **Selection:** We selected the top 1% of papers ranked by *bias span* ($Range_p$) for each conference year (2023, 2024, and 2025). This resulted in a dataset of 219 papers (34 from 2023, 71 from 2024, and 114 from 2025).

2. **Feature Extraction:** We aggregated all extracted pros and cons for these papers, resulting in a total of 5,186 distinct arguments (658 in 2023, 1572 in 2024, 2956 in 2025).

3. **Categorization:** We applied a lightweight taxonomy to classify each argument into one of six categories: *Novelty/Innovation*, *Evidence Strength*, *Completeness/Ablation*, *Clarity/Presentation*, *Broad Impact/Relevance*, and *Other*.

### D.2. Distribution of Disagreement Sources

We analyzed the distribution of argument categories to identify the primary drivers of divergence. Table 8 summarizes the results across the three years.

*Table 8.* Argument-category distribution in extreme-divergence papers

| Category | 2023 | 2024 | 2025 |
|---|---|---|---|
| Evidence Strength | 27.20% | 30.73% | 32.24% |
| Other | 29.48% | 27.35% | 26.15% |
| Clarity/Presentation | 20.82% | 17.18% | 15.97% |
| Broad Impact/Relevance | 12.77% | 10.81% | 11.91% |
| Novelty/Innovation | 7.75% | 9.29% | 7.78% |
| Completeness/Ablation | 1.98% | 4.64% | 5.95% |

"Evidence Strength" is consistently the most frequent category (approx. 30%), indicating that disagreements often hinge on the validity of experiments or benchmarks. "Clarity/Presentation" also plays a significant role, particularly in 2023 (20.82%), suggesting that writing quality acts as a significant differentiator.

### D.3. Weight Analysis by Category

To quantify the impact of different types of arguments on the final score, we calculated the average weight assigned to pros and cons within each category. The results are presented in Table 9.

*Table 9.* Average rationale weights by category

| Category | Mean Pro Weight | Mean Con Weight |
|---|---|---|
| Novelty/Innovation | +1.50 | -0.95 |
| Broad Impact/Relevance | +1.32 | -0.92 |
| Evidence Strength | +1.28 | -0.97 |
| Completeness/Ablation | +1.27 | -0.91 |
| Other | +1.25 | -0.91 |
| Clarity/Presentation | +0.95 | -0.81 |

Notably, arguments related to "Novelty/Innovation" receives the highest positive weights on average (+1.50), confirming that perceived originality is a strong driver of high scores. Conversely, "Evidence Strength" failures incurred the strongest penalties (-0.97), highlighting the critical importance of rigorous validation.

# E. Experiment 4: Intervention Study Details

This appendix provides the experimental protocols, prompt templates, and detailed case studies for the "LLM-as-a-Reviewer" intervention simulation (Experiment 4).

## E.1. Experimental Design

The intervention study simulates a feedback loop where reviewers are alerted to significant discrepancies between their numerical scores and their written critiques. The procedure consists of three stages:

**1. Sample Selection**    We selected a random sample of $N = 200$ reviews from the ICLR 2023-2025 dataset that exhibited a high absolute residual ($|r_u| > 2.0$). These represent cases where the score is significantly disconnected from the mechanical anchor derived from the text.

**2. Proxy Reviewer Simulation (Intervention)**    We employed GPT-4-turbo as a "Proxy Reviewer" to simulate the original human reviewer. The model was provided with:

- The original paper title and abstract.

- The original review text (strengths and weaknesses).

- The original score ($s_u$) and the mechanical anchor score ($s_{anchor}$).

- A prompt asking it to reflect on the discrepancy (see Section E.2).

The proxy reviewer was given two options:

1. **Option A (Revise)**: Acknowledge the discrepancy and provide a revised score ($s_{new}$) that better aligns with the text.

2. **Option B (Justify)**: Defend the original score by providing additional context or implicit reasoning that the mechanical anchor missed.

**3. Validation and Re-scoring**

- If the proxy selected **Option A**, we calculated the new residual $|s_{new} - s_{anchor}|$.

- If the proxy selected **Option B**, we fed the new justification into the Anchor Scoring Model (GPT-4o-mini) to compute a re-adjusted anchor $s'_{anchor}$ and updated residual $|s_u - s'_{anchor}|$.

## E.2. Prompt Templates

The following prompt was used to present the discrepancy to the Proxy Reviewer.

---

**SYSTEM_PROMPT = """**
*You are an experienced academic reviewer. You wrote the following review and gave this paper a score of \*\*{original_score} out of 10\*\*.*

**Context:**
As part of a routine quality check, your review has been analyzed. Based on the explicit strengths and weaknesses you mentioned, a mechanical calculation suggests a score around \*\*{anchor_score}\*\*.

**Note:** This is a simple mechanical analysis that may or may not align with your actual judgment. The discrepancy is \*\*{mismatch:.1f} points\*\*.

—

**Paper Information:**
Title: {title}

Abstract: {abstract}

---

**Your Original Review:**
{review_text}

**Key Points You Mentioned:**

Strengths:
{pros_text}

Weaknesses:
{cons_text}

—

**Your Task:**
Reflect on your evaluation. Consider:

1. **Review your written critique**: Does the tone and content of your review reasonably support the score of {original_score}?

2. **Consider the paper's merit**: Based on the actual paper quality, does your score of {original_score} fairly represent your assessment?

3. **Implicit factors**: Are there unstated considerations (writing quality, significance, rigor, or subjective judgments) that influenced your score but weren't explicitly mentioned?

**Two Options:**

**Option A - Revise Score**: If, upon reflection, you believe your score doesn't accurately reflect your written assessment or the paper's actual quality, provide a revised score.

**Option B - Justify Score**: If your score is appropriate and reflects factors beyond what was mechanically extracted, explain why your original score is justified.

**Important:** Most experienced reviewers have well-calibrated scoring. However, occasional mismatches between written reasoning and scores do occur. Be objective in your self-assessment.

**Respond in JSON format:**
{{
"decision": "A" or "B",
"new_score": (integer 1-10, only if decision is A, otherwise null),
"explanation": "Your reasoning (2-3 sentences)"
}}

The mechanical calculation of {anchor_score} is provided for reference only. Your decision should reflect your genuine professional judgment after re-examining the paper and your review.
"""""

*Figure 18.* Prompt used for the proxy-reviewer intervention

### E.3. Case Study

We illustrate the intervention with a real example from the simulation.

**Case 195 (ICLR 2025)**

- **Original Score**: 5.0

- **Anchor Score**: 9.0 (Strong Accept range based on text)

- **Initial Residual**: 4.0 (Huge discrepancy)

- **Review Text**: The reviewer listed "novelty, theoretical guarantees, strong experimental results" as strengths, with only minor weaknesses like "lack of implementation details."

**Proxy Reviewer Response (GPT-4.1):**

"*Upon reflection, my written review highlights several notable strengths... While I did note some weaknesses regarding practical implementation... these do not appear severe enough to warrant a score as low as 5.0... A score of 7 more accurately reflects a solid paper with some limitations.*"

**Outcome**: Use of Option A. The score was revised from $5.0 \rightarrow 7.0$. The residual dropped from $4.0 \rightarrow 2.0$.

### E.4. Quantitative Summary

Table 10 summarizes the aggregated results of the intervention on the $N = 200$ high-residual cases.

*Table 10.* Impact of the post-check intervention

| Metric | Pre-Intervention | Post-Intervention | Change |
|---|---|---|---|
| Mean Absolute Residual | 2.93 | 1.06 | -1.87 ($p < 10^{-70}$) |
| Compliance Rate (Flip Rate) | - | 76.5% | - |
| Justification Rate | - | 23.5% | - |

### E.5. Cross-Family Check for Evidence 4 Post-Check Simulation

**Cross-family proxy reviewer.** The post-check simulation may suffer from model circularity if the proxy reviewer and anchor are from related model families. We therefore replace the proxy reviewer with Qwen-2.5-72B-Instruct while keeping the anchor model, sample set, and metrics unchanged. The Qwen run uses exactly the same post-check prompt, input fields, Option A/B response format, parsing rule, and re-scoring procedure as the original proxy-reviewer simulation in Appendix E.2; only the proxy reviewer model is changed. The residual reduction remains comparable to the original setting.

*Table 11.* Cross-family validation for the post-check simulation

| Metric | Original Proxy Reviewer | Qwen Proxy Reviewer |
|---|---|---|
| Total cases | 200 | 200 |
| Anchor model | GPT-4o-mini | GPT-4o-mini |
| Proxy reviewer | GPT-4-turbo | Qwen-2.5-72B-Instruct |
| Decision A count | 153 | 130 |
| Decision B count | 47 | 70 |
| Flip rate | 0.765 | 0.650 |
| Baseline residual mean | 3.1285 | 3.1285 |
| Intervention residual mean | 1.2545 | 1.3505 |
| Residual reduction | 1.8740 | 1.7780 |

## F. Experiment 5: Borderline Paper Analysis Details

This appendix provides the detailed methodology and data sources for the borderline paper citation analysis presented in Section 4.5.

### F.1. Experimental Procedure

The borderline analysis was conducted to validate whether anchor scores identify higher-quality papers within the same nominal score band. We focus on the borderline zone (scores 4–6), where accept/reject decisions are most contentious and calibration matters most.

**Data Sources.** We analyzed ICLR 2023 ($N = 3,507$ papers) and ICLR 2024 ($N = 7,150$ papers) submissions with available review data and citation counts. The 2025 cohort was excluded from this analysis, as the insufficient post-publication window renders citation counts too sparse to serve as a reliable proxy for scientific impact. Citation counts were obtained from Semantic Scholar API as of January 2026, providing approximately 2.5 years of citation accumulation for ICLR 2023 papers and 1.5 years for ICLR 2024 papers.

**Score Band Partitioning.** For each paper, we computed:

- Average reviewer score: $s_p^H = \frac{1}{|\mathcal{R}(p)|} \sum_{j \in \mathcal{R}(p)} s_{p,j}^H$

- Average anchor score: $s_p^A = \frac{1}{|\mathcal{R}(p)|} \sum_{j \in \mathcal{R}(p)} s_{p,j}^A$

Papers were then partitioned into score bands (4–5, 5–6) based on these averages under each scoring regime separately.

**Citation Metrics.** For each score band under each regime, we computed:

- Mean citation count

- Median citation count

- Total citation count

- Citation efficiency: $\frac{\text{Citation share}}{\text{Paper share}}$

## F.2. Aggregated Results

Table 12 summarizes the citation statistics for borderline papers across both years and scoring methods.

*Table 12.* Borderline paper citation statistics

| Year | Band | Method | Papers | Mean Cites | Improvement |
|------|------|--------|--------|-----------|-------------|
| 2023 | 4–5 | Reviewer | 687 | 13.6 | +53% |
| | | Anchor | 961 | 20.8 | |
| | 5–6 | Reviewer | 1,019 | 22.7 | +94% |
| | | Anchor | 1,393 | 44.0 | |
| 2024 | 4–5 | Reviewer | 1,834 | 13.1 | +77% |
| | | Anchor | 1,679 | 23.2 | |
| | 5–6 | Reviewer | 1,957 | 29.1 | +19% |
| | | Anchor | 2,835 | 34.5 | |

## F.3. Citation-Analysis Leakage Control

**Citation-analysis leakage control.** Because citation outcomes may create hindsight leakage concerns, we rerun the borderline citation analysis with gpt-3.5-turbo-1106 on ICLR 2023–2024 and also evaluate ICLR 2025 using gpt-4o-mini. The score-band construction, citation source, citation aggregation, and reviewer-versus-anchor comparison are identical to the main borderline analysis in Appendix F; only the anchor model or dataset changes. The anchor-selected borderline groups continue to show higher mean citation counts than reviewer-score-selected groups.

*Table 13.* Borderline citation analysis with an older model on ICLR 2023–2024

| Year | Score Band | Reviewer Mean Cit. | Anchor Mean Cit. | Improvement |
|------|-----------|--------------------|--------------------|-------------|
| 2023 | 4–5 | 13.61 | 18.07 | +32.8% |
| 2023 | 5–6 | 22.67 | 43.35 | +91.2% |
| 2024 | 4–5 | 13.10 | 19.49 | +48.8% |
| 2024 | 5–6 | 29.06 | 32.20 | +10.8% |

*Table 14.* Borderline citation analysis on ICLR 2025 using gpt-4o-mini

| Score Band | Reviewer Mean Cit. | Anchor Mean Cit. | Improvement |
|-----------|--------------------|--------------------|-------------|
| 4–5 | 12.53 | 14.02 | +11.9% |
| 5–6 | 15.14 | 20.40 | +34.8% |

## F.4. Boundary-Focused Validity Checks

**Robustness to unmeasured confounding.** We add an E-value analysis to quantify how strong an unmeasured confounder would need to be to explain away the observed relationships. The reported E-values follow the sensitivity-analysis definition of VanderWeele and Ding (VanderWeele & Ding, 2017): an unmeasured confounder would need to be associated with both the exposure and outcome by at least the reported risk-ratio scale, conditional on measured quantities, to explain away the effect. Because our outcomes are continuous, we first convert each effect to a standardized effect size and then use the standard continuous-outcome approximation $RR \approx \exp(0.91|d|)$, followed by $E = RR + \sqrt{RR(RR - 1)}$. For absolute bias magnitude and within-paper bias span, $d$ is the mean divided by its empirical standard deviation; for the anchor–actual score association, we convert Pearson $r$ to $d = 2r/\sqrt{1 - r^2}$. The 95% CI E-value uses the effect-size limit closest to zero. As a diagnostic, the implementation also computes one-sample, directional, per-paper, score-segment, and sequential e-values for the bias distribution; Table 15 reports the three sensitivity checks most directly tied to the main residual-based claims. The values below indicate that the main effects would require strong unmeasured confounding to disappear.

*Table 15.* E-value robustness analysis for residual-based effects

| Analysis | $n$ | Effect Size | E-value | E-value 95% CI |
|----------|-----|-------------|---------|----------------|
| Absolute bias magnitude $|bias|$ | 27,601 | $d = 1.2910$ | 5.9290 | 5.8593 |
| Anchor score vs. actual score correlation | 27,601 | $r = 0.4778, d = 1.0879$ | 4.8248 | 4.7007 |
| Within-paper bias span | 7,150 | $d = 2.1212$ | 13.2631 | 12.9749 |

**Boundary-focused causal-style checks.** We further use a regression discontinuity design around the accept/reject boundary and a mediation analysis to probe whether miscalibration behaves like independent noise rather than a paper-quality proxy (Imbens & Lemieux, 2008; Lee & Lemieux, 2010; Baron & Kenny, 1986; Imai et al., 2010). The RDD estimates a positive citation gain from acceptance near the boundary, while the interaction between treatment and residual magnitude is negative in ICLR 2024. The mediation analysis shows that paper quality does not significantly predict miscalibration, while miscalibration remains negatively associated with citation outcomes after controlling for quality. These analyses should be read as causal-style robustness checks over proxy outcomes, not as direct proof of improved review quality.

**RDD design.** The RDD is run on paper-level ICLR 2023–2024 data after merging average reviewer scores, average anchor scores, residual-dispersion statistics, accept/reject decisions, and citation counts. The running variable is the average actual reviewer score centered at an empirically estimated accept/reject cutoff $c_y$ for each year, and treatment is $T = \mathbb{1}[\bar{s}_p^H \geq c_y]$. We estimate the local linear model

$$Y_p = \alpha + \tau T_p + \beta_1 X_p + \beta_2 (T_p X_p) + \epsilon_p$$

where $X_p = \bar{s}_p^H - c_y$ and $Y_p = \log(1 + \text{citations}_p)$. The main RDD table uses bandwidth 1.0 around the cutoff and reports $\tau$ as the local average treatment effect; standard errors and $p$-values are obtained from 2,000 bootstrap resamples with a fixed

random seed. We also run bandwidth sensitivity checks over $\{0.5, 0.75, 1.0, 1.25, 1.5, 2.0\}$ and a simplified McCrary-style density diagnostic within bandwidth 2.0 using 5,000 permutations (McCrary, 2008).

For the residual-moderator test, we use bandwidth 1.5, standardize paper-level mean absolute residuals, and fit

$$Y_p = \alpha + \tau T_p + \beta X_p + \gamma Z_p + \eta(T_p X_p) + \delta(T_p Z_p) + \epsilon_p$$

where $Z_p$ is the standardized mean absolute residual. The reported moderator coefficient is $\delta$; a negative value means that larger miscalibration attenuates the citation gain associated with acceptance near the boundary.

**Mediation design.** The mediation analysis uses $X =$ standardized average anchor score as a rationale-implied paper-quality proxy, $M =$ standardized mean absolute residual as miscalibration, and $Y = \log(1 + \text{citations})$. We apply the Baron–Kenny decomposition by fitting three linear models: $X \to Y$ for the total effect $c$, $X \to M$ for the $a$ path, and $(X, M) \to Y$ for the $b$ path and direct effect $c'$. We compute the indirect effect $a \times b$, run a Sobel test (Sobel, 1982), and estimate bootstrap confidence intervals for indirect, direct, and total effects using 5,000 resamples. The implementation also includes a rho-based sensitivity analysis over $\rho \in [-0.9, 0.9]$, following the sequential-ignorability diagnostic logic of causal mediation analysis (Imai et al., 2010).

*Table 16.* Main RDD effect near the accept/reject boundary

| Year | Cutoff | LATE | SE | $p$-value | $N$ (left/right) |
|------|--------|------|------|-----------|------------------|
| 2023 | 5.62 | 0.498 | 0.141 | 0.0004 | 961 / 1007 |
| 2024 | 5.83 | 0.219 | 0.107 | 0.040 | 1952 / 1552 |

*Table 17.* Miscalibration moderator in the ICLR 2024 RDD analysis

| Variable | Coefficient | SE | $p$-value |
|----------|-------------|------|-----------|
| Treated $\times |residual|$ | -0.119 | 0.044 | 0.007 |

*Table 18.* Mediation analysis on ICLR 2024 ($N = 7{,}150$)

| Path | Coefficient | $p$-value |
|------|-------------|-----------|
| $a$: quality $\to$ miscalibration | 0.015 | 0.199 |
| $b$: miscalibration $\to$ citations, controlling for quality | -0.065 | 0.001 |
| Indirect $a \times b$ | -0.001 | 0.231 |

# G. Usage of LLMs

We used GPT-5.2 for language polishing (grammar, clarity, and concision) and code implementation in our experiments.

