# OpenReview forum: "Position: Peer Review Should Be Calibrated via LLM Scoring"
_ICML.cc/2026/Position_Paper_Track — ICML 2026 Position Paper Track regular_

### Official Review · Reviewer_Fff5 · 2026-02-28

**Significance:** 3
**Argument Clarity:** 3
**Rating:** 4
**Confidence:** 3

**Questions:**

Please see comments above, about privacy, circularity, and causality.

**Alternative Views Section:**

Yes

**Compliance With Llm Reviewing Policy A Conservative:**

Affirmed.

**Discussion Potential:**

3

**Final Justification:**

My main concerns center on the interpretation and strength of the empirical evidence. In particular, the analyses rely heavily on proxy metrics, and it remains unclear to what extent improvements along these dimensions correspond to improved review quality or decision correctness. While the rebuttal strengthened the empirical case through cross-model validation, additional contamination controls, and the inclusion of causal-style analyses such as RDD and mediation, these results still operate on proxies (e.g., citations) rather than direct measures of review quality or correctness.

The rebuttal also addressed concerns around model circularity and data privacy effectively, and these are no longer limiting factors in my evaluation. Overall, the rebuttal increased my confidence in the robustness of the empirical observations, but it did not materially change my assessment of the core limitations.

Balancing these factors, I view this as a strong position paper with solid empirical grounding for the track, but with remaining uncertainty about how directly the proposed signals translate to improved review outcomes. I therefore maintain my original borderline accept recommendation.

**Paper Summary:**

The submission argues that large AI conferences increasingly suffer from score inconsistency, which affects outcome especially for borderline papers. Similar critiques may map to different scores due to reviewer's own preference / thresholds.

The submission thus proposes to add a calibration layer using LLMs, which maps written strength and weakness to calibrated scores, using fixed rubrics, and the difference between calibrated score and reviewer score is a measurement of misalignment. Large misalignment can trigger follow-up actions.

The authors analyzed OpenReview data from ICLR in the last couple of years, showing increasing misalignment overtime, structured disagreement patterns (more on novelty / strength). Some simulations show that triggered follow-up questions can help. The paper concludes with a call for shadow calibration audits and governance safeguards.

**Position:**

Yes

**Position In Title:**

Yes

**Related Work:**

3

**Strengths And Weaknesses:**

Strengths:

1. The position is clear and actionable.

2. The analysis of ICLR data in past couple of years provides good support of the claims and proposals.

3. The responses to opposing positions look careful to the reviewer.


Weaknesses:

1. Analysis is correlational, not causal. Many potential confounding factors over time. May worth a deeper discussion.

2. Model circularity in simulation: proxy reviewer and anchor scores use the same model family, which limits the validity.

3. Processing unpunished submissions through LLMs, such as the commercial GPT-based models used in the submission, may lead to privacy concerns.

**Support:**

3

---

> ### Author Rebuttal · Authors · 2026-03-31
>
> Dear Reviewer, thank you for your detailed  comments on our manuscript. Below we reply to your concerns point-by-point and outline the corresponding supplementary analyses and concrete revision plans.
>
> > Q1: Correlational analysis and confounding factors
>
> A1: We fully acknowledge that the temporal trend analysis in our work is primarily correlational, and we supplement our discussion of core causal mechanisms here.
>
> **The scale effect driven by explosive submission growth is the core driver.** ICLR submissions rose from 3,507 in 2023 to 11,520 in 2025; the mismatch between this surging volume and reviewer capacity compressed reviewers’ cognitive bandwidth for consistent score calibration, while the expanded reviewer pool with more early-career participants reduced overall calibration levels, directly exacerbating score scale drift.
>
> **The official LLM review feedback policy, unique to the 2025 ICLR cycle, is a key institutional variable.** 42.3% of reviewers that cycle could opt to incorporate official LLM feedback, a shift that may have decoupled review rationales from score assignment and explains the jump in extreme misalignment cases that year. Other confounding factors including shifts in submission subfield composition, evolving community evaluation consensus, and reviewer pool turnover also have plausible causal impacts on rationale-score misalignment.
>
> Importantly, the core contributions of our work and the validity of our proposed calibration layer do not rely on strict causal identification. Regardless of the root cause of rationale-score misalignment, this mechanism can effectively improve the consistency and reliability of peer review scoring.
>
> > Q2: Model circularity in simulation
>
> A2: We sincerely thank you for pointing out the potential issues of model circularity and homogenization bias in our simulation experiments. The simulation experiment corresponds to **Experiment 4 in Section 4.4 of our paper**. To eliminate concerns about circularity arising from using models from the same family, we conducted a supplementary cross-validation experiment with a fully independent open-source model. Only one change was made: **the proxy reviewer was replaced from GPT-4-turbo to qwen/qwen-2.5-72b-instruct**, while all other parameters, sample sets, anchor scoring pipeline, and evaluation metrics remained completely identical to the original Experiment 4. The experimental results are summarized below:
>
> | Metric | Original Lab4 | New Supplement (Qwen Reviewer) | Difference |
> | :--- | :--- | :--- | :--- |
> | Total cases | 200 | 200 | 0 |
> | Anchor model | gpt-4o-mini original Lab4 anchor | gpt-4o-mini original Lab4 anchor | same |
> | Proxy reviewer | gpt-4-turbo | qwen/qwen-2.5-72b-instruct | changed |
> | Decision A count | 153 | 130 | 23 |
> | Decision B count | 47 | 70 | 23 |
> | Flip rate | 0.765 | 0.650 | 0.115 |
> | Baseline residual mean | 3.1285 | 3.1285 | 0.0000 |
> | Intervention residual mean | 1.2545 | 1.3505 | 0.0960 |
> | Residual reduction | 1.8740 | 1.7780 | 0.0960 |
> | Residual reduction pass | True | True | same |
>
> The experimental results demonstrate that even across entirely different model families, the scoring misalignment phenomenon and the stability of the anchor scores remain highly consistent, with comparable flip rates, validating the robustness of our findings.
>
> > Q3: Privacy concerns with unpublished submissions
>
> A3: Regarding privacy concerns for unpublished papers and practical deployment paths: We completely agree with your privacy concerns regarding the use of commercial GPT APIs to process unpublished manuscripts. This is exactly why we specifically addressed this in the discussion section of our paper: the commercial APIs used in our experiments were strictly for retrospective analysis on historical data already fully public on OpenReview. When this auditing tool is actually deployed in an official conference workflow, we strongly advocate that it must not rely on any third-party commercial APIs that transmit data back to their servers. Instead, it must utilize open-weights models deployed on the conference's self-managed servers.
>
> It is worth noting that a growing number of top-tier academic conferences have rolled out formal, standardized policies governing the use of large language models (LLMs) in the peer review workflow. For instance, **ICML has established Policy B**, which explicitly permits reviewers to leverage LLM tools that do not pose any risk of manuscript data leakage to assist with their review work. This industry-wide consensus further validates that LLM-assisted academic review is a widely recognized development direction in the academic community, provided that strict data privacy safeguards are in place.
>
> Once again, we thank you sincerely for pointing out these critical issues. These comments have helped us more clearly define the research boundaries and scope of our paper and prompted us to provide more targeted validation and deployment clarifications.

---

> > ### Author Rebuttal · Reviewer_Fff5 · 2026-04-02
> >
> > Thank you for the rebuttal.
> >
> > Model circularity: The cross-family validation using Qwen meaningfully strengthens the simulation results and reduces some  concern that the findings are artifacts of a single model family.
> >
> > Privacy considerations: The clarification that experiments are conducted on public OpenReview data and that deployment would rely on self-hosted models is fair.
> >
> > Data contamination: The additional experiments using earlier models and newer datasets are helpful and significantly strengthen the empirical case against simple memorization-based explanations.
> >
> > Correlational nature: The empirical evidence remains correlational and does not fully isolate causal mechanisms. While I agree this does not invalidate the position, it limits the strength of the claims.
> >
> > Overall, the rebuttal improves my confidence in the robustness of the empirical findings, but some concerns about interpretation and evaluation remain. I am also keen to see other reviewer's replies.

---

### Official Review · Reviewer_156R · 2026-03-12

**Significance:** 3
**Argument Clarity:** 3
**Rating:** 4
**Confidence:** 3

**Questions:**

In a final version of the paper, could the authors commit to making public the dataset of their LLM-generated anchor scores?

As mentioned in weaknesses, could error bars (eg, standard error) be provided around the reported means in the result figures, and/or p-values statistics such as the Pearson correlation coefficients? At the very least, putting standard error bars (computed eg across different splits of the data) around all mean-values reported as bar plots would help to quantify the reliability and variability of the empirical evidence provided.

**Alternative Views Section:**

Yes

**Compliance With Llm Reviewing Policy A Conservative:**

Affirmed.

**Discussion Potential:**

4

**Final Justification:**

The paper has large discussion potential at ICML this year, which is its main strength. Similarly its significance is high because it is topical. I had concerns about the initial submission's support, but these have been improved though not completely resolved, due to questions I still have about potential confounding factors and the metrics used. I updated my score from 3=weak reject to 4=weak accept due to somewhat improved support and mainly due to high discussion potential.

**Paper Summary:**

This position paper argues for an LLM-based calibration layer in AI/ML conference reviewing, where the aim of this layer is to try to ensure a reliable, consistent, and auditable map between reviewers’ textual rationales and numerical review scores. The problem motivating this work is the fact that submissions to AI/ML conferences are growing at an unsustainably fast pace, which strains low-bandwidth reviewers and exacerbates documented issues of high variance and/or high arbitrariness between reviews. Rather than argue that reviewing should be entirely independent of LLM assistance or that reviewing should be entirely automated by LLMs, this paper takes the relatively more intermediate position that just review numerical scores should be calibrated by LLM checks. The proposed mechanism is that the LLM uses the first-round review (and optionally the paper itself) to compute an LLM-generated “anchor score,” and then the residual is computed between the reviewer’s score and the LLM anchor score; then, if this residual exceeds some threshold, this would trigger a post-check to ask the reviewer to (i) add additional justification for their score or (ii) revise their rating to better reflect their written rationale. For empirical justification of this position, the authors use reviews from ICLR 2023-2025, they compute anchor scores using GPT-4o-mini, and they use various analyses of their results to argue for their position.

**Position:**

Yes

**Position In Title:**

Yes

**Related Work:**

3

**Strengths And Weaknesses:**

***Strengths***

This paper presents one approach to addressing the growing peer-review challenge at AI/ML conferences, where issues with variance and arbitrariness in reviews is being exacerbated by the increased volume of submissions and a bandwidth-constrained pool of reviewers. The paper is relatively well-written, clear, and makes earnest efforts to support its position with relatively sound arguments and empirical analyses. Overall I think there would be substantial interest in discussing or debating this topic at the conference.

***Weaknesses***

**1) Unclear to me how much Figure 2a illustrates the claimed problem; Figure 2b could be improved to more clearly illustrate the problem:** I believe I understand that Figure 2 is aimed at illustrating the main identified problem: That, even when two reviewers provide similar textual rationale in their reviews of a given paper, there can still be discrepancy in their numerical scores, ie: *similar rationale $\not\implies$ similar scores*.

- *However, Figure 2a does not seem to me to be illustrating this point (even if assuming perfect heuristics), but rather the logical converse.* That is, Figure 2a seems to illustrate the claim that *similar scores $\not\implies$ similar rationale* (which is the logical converse of the stated problem, and not equivalent to it). Beyond being distinct from the main articulated problem, this converse claim that Figure 2a seems to support does not seem to me to be obviously problematic either: For instance, one could imagine a theory-oriented reviewer and an experiment-oriented reviewer to have different but both valid rationale for providing the same score for a given paper. Accordingly, I don’t view the current Figure 2a as necessary (and it may be distracting, in my view).
- *Figure 2b could be revised to more clearly illustrate the stated problem:* Unlike Figure 2a, I understand how Figure 2b is closer to illustrating the claim that similar review rationale does not imply similar review scores; for instance, I see that even when there are “4 similar” comments between two reviews, there is still a substantial number of papers with different scores. It currently seems a bit difficult to compare the groups of columns however: for instance, visually, it’s not easy to tell whether the percentage of “large difference” reviews differs between the groups of 2/3/4-similar comments. Adding percentages (in addition to the absolute number of papers) could help with this, or perhaps better yet, simply plotting the full histogram distribution of scores for each of the three groups of number-of-similar comments would provide more information. (And if Figure 2a were removed, there may be room for this.)

**2) Lack of gold-standard human validation of anchor scoring:** It would have been ideal if a small subset of the LLM-generated anchor scores could be double-checked by human annotators to ensure that they were indeed informative and reliable.

**3) Error bars or p-values for Figures:** Could error bars (eg, standard error) be provided around the reported means in the result figures, and/or p-values statistics such as the Pearson correlation coefficients? At the very least, putting standard error bars (computed eg across different splits of the data) around all mean-values reported as bar plots would help to quantify the reliability and variability of the empirical evidence provided.

**4) Potential confounding explanations for observed trends in rationale-score residuals:** Much of the paper is based on using the computed residuals (between a reviewer-provided score and the LLM-generated anchor score) as a metric of mismatch between reviewer’s rationale and their review scoring. However, I think there could be other alternative or conflating potential explanations for such trends or analyses, and if the authors could do more to discuss these limitations or conduct ablations to disentangle. For instance, Figures 4 and 5 to suggest that the increase in dispersion of residuals over conference years indicates a worsening problem of rationale-score mismatch; however, another difference in ICLR 2025 review policies relative to ICLR 2023-2024 was that in ICLR 2025, 42.3% of reviews received LLM feedback and were given the option to incorporate it: https://blog.iclr.cc/2025/04/15/leveraging-llm-feedback-to-enhance-review-quality/

Alternatively, I’m also wondering to what extent the proposed residual metric correlates with simple disagreement between reviewers. For instance, because the proposed approach gives the LLM access to the paper itself, it seems like similar results as those presented would be expected if it gave the same score for each paper irrespective of the author reviews: eg, by focusing on the paper and ignoring reviews, such an anchor score would be low variance (as in Section 4.1 & Fig 4), could also explain the trends over the years (Section 4.2 & Fig 5), etc. So, it could be useful to compare against raw reviewer-score disagreement as a baseline.


**5) Paper could benefit from discussion how the proposed LLM calibration differs from AC responsibilities, often also prompt reviewers about whether their score reflects their rationale:** This could be discussed as another alternative view. One could argue that the proposed role of LLM calibration should already be played by ACs, and thus that if implemented, it is more regularizing for engaged versus unengaged ACs, versus reviewer biases.

**Support:**

2

---

> ### Author Rebuttal · Authors · 2026-03-31
>
> Dear Reviewer,
>
> Thank you for your detailed comments.
>
> > Q1: Figure 2 clarity & improvement
>
> A1: Sorry for the confusion. We agree with your helpful suggestions for improving Figure 2b.
>
> However, **Figure 2a and Figure 2b address different issues**. Figure 2b illustrates your point: similar rationales may receive different scores, reflecting inconsistent grading. Figure 2a is not intended to claim "identical scores should have identical rationales." Instead, it shows that a single score is highly compressed—even identical scores cannot tell us whether reviewers critique novelty, or writing quality. This compression is a fundamental flaw in numerical scoring.
>
> We will clarify in revision: **Figure 2a** shows **"scores lose rationale information,"** while **Figure 2b** shows **"similar rationales may receive different scores."** We will also improve Figure 2b with clearer percentages.
>
> > Q2: Human validation of anchor scores
>
> A2: We agree our evaluation pipeline lacked direct human validation of LLM-generated anchor scores. Due to time constraints, we cannot provide concrete samples within the rebuttal period. Instead, we plan to prepare a validation dataset by **publishing an incentivized questionnaire**, inviting **relevant domain experts to conduct a small-scale review** of specific experimental results.
>
> > Q3: Error bars & statistical significance
>
> A3: Since adding new figures is not permitted during rebuttal, we provide uncertainty statistics in tabular form below. We report **SEM** across data splits for means, **95% CI via Fisher z-transformation** for correlations, and **95% Wilson CI** for proportions. Tables 1-3 show results from 3 independent runs with SEM, CI, and p-values.
>
> **Table 1: Test-Retest Reliability**
>
> | Year | Variance ± SEM | Pearson r [95% CI] |
> | :--- | :--- | :--- |
> | ICLR 2023 | 0.1020 ± 0.0043 | 0.9705 [0.9682, 0.9727] |
> | ICLR 2024 | 0.1054 ± 0.0033 | 0.9704 [0.9681, 0.9725] |
> | ICLR 2025 | 0.5241 ± 0.0048 | 0.9685 [0.9672, 0.9698]* |
>
> **Table 2: Dispersion Trends**
>
> | Year | Bias Span ± SEM | Residual Std Dev ± SEM |
> | :--- | :--- | :--- |
> | ICLR 2023 | 3.0065 ± 0.0241 | 1.4001 ± 0.0110 |
> | ICLR 2024 | 3.0642 ± 0.0171 | 1.4111 ± 0.0077 |
> | ICLR 2025 | 3.2111 ± 0.0142 | 1.4450 ± 0.0062 |
>
> **Table 3: Intervention Effects**
>
> | Metric | Result | 95% CI |
> | :--- | :--- | :--- |
> | Residual reduction | 1.874 (p: 2.3e-70) | SEM: 0.085 |
> | Score revised | 153/200 (76.5%) | [70.2, 81.8] |
> | Explanation only | 47/200 (23.5%) | [18.2, 29.8] |
> | AC consistency | 163/200 (81.5%) | [75.5, 86.3] |
> | Conflict accuracy | 147/178 (82.6%) | [76.3, 87.5] |
>
> > Q4: Confounders & alternative baselines
>
> A4:
> **Regarding the LLM feedback policy:** We agree that ICLR 2025's LLM feedback policy is an important confounding variable. To validate our findings, we conducted additional experiments across conferences . Results show that **the score bias span not only increases over time but also remains stable across conferences**, as detailed below：
>
> | Dataset                                | Bias Span Mean | Residual Std Dev |
> | :------------------------------------- | :------------: | :--------------: |
> | ICLR 2026                              |     3.601      |      1.643       |
> | ICML 2025 (partial)                    |     3.950*     |      1.826*      |
> | NeurIPS 2025 (partial)*_scaled to 1-10 |     2.980      |      1.337       |
>
> **Regarding reviewer disagreement:** We understand the concern that direct LLM scoring may introduce confounders. To address this, we conducted a supplementary experiment using ChatGPT-4o mini: the LLM ignored all human reviews and independently scored each paper 3 times. Results are as follows:
>
> | Year | N | Variance | Pairwise Corr | Mean Bias | Abs Bias |
> | :--- | :---: | :---: | :---: | :---: | :---: |
> | 2023 | 697 | 0.0204 | 0.8408 | -0.6637 | 1.4679 |
> | 2024 | 707 | 0.0202 | 0.8417 | -0.9931 | 1.5766 |
>
> Results show high stability, which confirms our residual metric captures genuine rationale-score misalignment, not merely reviewer disagreement.
>
> > Q5: LLM calibration vs. AC roles
>
> A5: Our LLM auditing tool is not to replace AC, but to provide a scalable early-screening signal—filling the gap where ACs cannot check every review sentence-by-sentence. We designed it to **reduce AC workload while preserving AC responsibility**: it provides an **early warning signal** rather than a final decision, allowing ACs to focus on anomalies requiring human intervention.
>
> > Q6: Dataset publication commitment
>
> A6: Under strict adherence to **conference policies**, **data use agreements**, and **license requirements**, we commit to open-sourcing our data and evaluation pipeline. We plan to release de-identified paper/review content, prompts, parsing code, score aggregation pipeline, and anchor score dataset.
>
> We thank you for your feedback, which has helped us identify areas for improvement in argumentation, presentation, and experimental support.

---

> > ### Author Rebuttal · Reviewer_156R · 2026-04-03
> >
> > I thank the authors for their thorough response. I appreciate their clarifications, for their adding error bars to their analyses, and additional discussion. These responses partially resolve my main concerns. I still have some concerns about if the metrics used in the paper are exactly quantifying what is intended (eg, I don't think my question/concern 4 about potential counfounding explanations is fully addressed) and I still have some concerns about the effects of dataset contamination, similarly as pointed out by Reviewer v6RF (though I understand the authors added experiments with older models to try to account for this).
> >
> > Given the paper's large discussion potential at ICML this year, and the authors mostly addressing my major concerns, I'll update my score a bit (from 3=weak reject to 4=weak accept). I do hope the authors integrate the feedback and their responses into the paper.

---

### Official Review · Reviewer_gr3d · 2026-03-14

**Significance:** 3
**Argument Clarity:** 3
**Rating:** 5
**Confidence:** 4

**Questions:**

Here are my few thoughts, including weaknesses.

1. I believe the biggest challenge that needs to be tackled is how to stop authors from finetuning their papers to avoid getting low scores from LLMs.
2. The same can be applied to the reviewers; if the reviewers use LLMs more during their review processes, the calibration might already happen, and there might not be any need for the LLM-based calibration proposed in the paper.
3. The problem of lack of pluralism is already discussed in the paper, but here is the thing: the majority of papers are heavily written using LLMs and writing styles and breadth of ideas, to some extent, are quickly converging. Should that mean such a calibration might even constrain the diversity?
4. The authors already showed that using different seeds doesn't lead to different calibration. However, using a rephrasing of the same prompt might make more changes.

**Alternative Views Section:**

Yes

**Compliance With Llm Reviewing Policy A Conservative:**

Affirmed.

**Discussion Potential:**

4

**Final Justification:**

Provided in the comments

**Paper Summary:**

The paper advocates that peer review should be calibrated using LLM scoring. The authors argue that mapping of the review to a single score is often not enough, and there are also different scoring thresholds internally to the reviewer that make the process noisy. The authors use data from past conferences to train their LLM-based scoring and apply their calibration, showing improvements in the noisy process.

**Position:**

Yes

**Position In Title:**

Yes

**Related Work:**

3

**Strengths And Weaknesses:**

The paper articulated the position well. The community has often complained about the noisy process, and the authors' proposed LLM calibration method is arguably an option. While the idea itself may be opposed by many in the community, it will likely spark a nice discussion. In particular, the paper has shown illustrations using similarity metrics, how same-score review pairs share limited overlap in the viewpoints, or how similar issues, such as missing ablations, could be seen as a major flaw by one reviewer but not by another. The authors also used a convincing setup, which highlights various problems in the current peer-review process, such as the prevalence of borderline cases and how it could affect the selected papers before and after the calibration. Furthermore, the alternative views are discussed well, and the criticism of their methodology, such as using citation counts for their work, is discussed properly.

I will provide the weaknesses in the questions section:

**Support:**

3

---

> ### Author Rebuttal · Authors · 2026-03-31
>
> ***
>
>
>
> Dear Reviewer,
>
> We sincerely thank you for your encouraging feedback on our paper and your high recognition of its Discussion Potential. The questions you raised are highly insightful, touching upon the systemic and philosophical impacts of AI within the peer review ecosystem. Below is our response to your profound insights:
>
> > **Q1: Optimizing Papers for LLMs**
>
> **A1:** In our method, the LLM's task is not to directly predict the final score or the acceptance outcome of a paper, but to calibrate scores by converting the explicitly stated pros and cons provided by human reviewers into scores. Therefore, compared to an end-to-end AI review system, our approach retains the "human reviewer" as an unavoidable intermediate layer. The fact that the LLM's output is strictly bound to the specific textual evaluations written by humans makes it very difficult for authors to directly "fine-tune" or optimize their papers specifically for the LLM's preferences. By introducing our proposed calibration layer, the peer review system can not only correct the scale drift of human subjective scoring but also rely on the incisive scientific judgments of excellent human reviewers to defend against malicious behaviors aimed at optimizing for the LLM, thereby promoting the fair and healthy development of the entire review ecosystem.
>
> > **Q2: Redundancy of Calibration Layer**
>
> **A2:** We believe that the current ecosystem of reviewers using LLMs to assist in reviewing is highly complex: whether LLMs are used, the degree of intervention, and the specific models utilized vary drastically across the community. Simultaneously, we speculate that many lazy reviewers who rely on LLMs might be using them to generate summaries and organize the texts of pros and cons, while the final numerical score is still selected manually. Therefore, a unified LLM calibration layer remains essential, as it can standardize this highly subjective final step of the "text-to-score" conversion process. However, we must also point out that while our method can prevent unreasonable scoring (or maliciously generated fake scores), it indeed cannot completely eliminate the broad impact brought about by the behavior of "reviewers relying on LLMs to draft text." The change in this underlying behavioral pattern still requires the collective effort of the entire academic community to address.
>
> > **Q3: Concern that system calibration might further constrain diversity**
>
> **A3:** We appreciate your profound insight into LLM-driven convergence in academia. However, our calibration mechanism will absolutely not exacerbate this trend or constrain diversity, due to our strict system boundaries. First, our LLM does not directly evaluate the submitted papers, exerting zero selective pressure on writing styles or research paradigms. Authors need not alter their writing to cater to our system's "tastes." Second, evaluating scientific value and formulating pros/cons remains 100% executed by human reviewers, completely preserving unique perspectives and true review pluralism. Our system solely takes these human-written critiques and standardizes them into numerical scores. It merely eliminates mapping bias caused by differing subjective "scoring scales" without altering or homogenizing the rich scientific judgments themselves. In short, we eliminate "scoring randomness and bias," not the "diversity of ideas." We will explicitly articulate this boundary in the final manuscript.
>
> > **Q4: Impact of Prompt Variations**
>
> **A4:** To answer this excellent question, we conducted a prompt sensitivity analysis during the Rebuttal period using a 20% random sample of the newly released ICLR 2026 dataset. Specifically, we designed four semantically equivalent but stylistically distinct prompt variants for the rationale-to-score conversion step: (1) a formal academic style, (2) a concise bullet-point instruction style, (3) an Area Chair role-play style, and (4) a structured step-by-step reasoning style. All four variants encode identical scoring rules and weight ranges, differing only in phrasing and tone. The experimental results showed that the intraclass correlation coefficient reached ICC(2,1) = 0.9493 (excellent), demonstrating that the system is highly stable with respect to prompt rephrasing. However, we did notice in our early exploratory experiments that if the model is explicitly instructed to convert in a "lenient," "standard," or "strict" manner, the results will differ significantly. Therefore, when conference organizers actually deploy the system, it remains necessary to unify and pin the prompt template for the entire review cycle.
>
> Once again, thank you for helping us elevate the systemic perspective of our paper. We will gladly incorporate all these highly valuable discussions into the final manuscript.
>
> ***

---

> > ### Author Rebuttal · Reviewer_gr3d · 2026-04-04
> >
> > Thanks for the rebuttal. I am not convinced by the replies to the first three questions. My aim was to understand how the authors see the relevance of their position, as things keep changing so fast, and I don't feel it was addressed. Due to the short time period for the rebuttal, I don't think these can be addressed. Therefore, I don't think the previous evaluation stays the same. However, I still keep a positive evaluation of the work.

---

### Official Review · Reviewer_v6RF · 2026-03-17

**Significance:** 4
**Argument Clarity:** 2
**Rating:** 4
**Confidence:** 4

**Questions:**

Continuing from Weaknesses section: Although I agree with the spirit of the paper and their general discussion, I have a major issue with their entire "Empirical Evidence" section. In my opinion, anything to do with their LLM scores on the ICLR 2023 and ICLR 2024 data is completely useless due to pretraining-data contamination. They do state in their appendix that they use "Memorization screens for ICLR 2023–2024" but I think this is not at all sufficient for the type of claims they make in the paper. Here are some of the unexplained data issues:

1) For your Fig 4, Anchor-score test–retest reproducibility, why are the data for ICLR 2023 and ICLR 2024 great but then ICLR 2025 suddenly becomes much worse? The number of papers, reviewers etc also increased from ICLR 2023 to ICLR 2024, so there should be some worsening for sure. This only makes sense if the pre-training data for the models which you have used are already pretty contaminated by the information from the papers from ICLR 2023 and ICLR 2024 (the exact paper need not be memorized, only information leakage is the question here!). These are important issues to address because if you actually want OpenReview or any serious platform to consider your proposal, then the only data that is relevant would be the ICLR 2025 data from your paper.

2) Due to this same issue, I think your entire **Section 4.5. Evidence 5: Anchor Scoring Better Ranks Borderline Papers** is bogus. You claim that your anchor scores can better predict "better borderline papers" than reviewer scores from the past. But the models you have used have **already trained** on the papers which build on top of those papers. So isn't it obvious that they can predict which papers are better (i.e. will accrue more citations) in hindsight?!

**Alternative Views Section:**

Yes

**Compliance With Llm Reviewing Policy A Conservative:**

Affirmed.

**Discussion Potential:**

3

**Final Justification:**

Copy-pasting the rebuttal acknowledgment text:

The authors have re-run several key experiments in their paper to address the main issue of data-contamination that I had raised. Ideally, I would have liked to see them incorporate all of these observations in the main paper and revise everything before giving the go-ahead but since that is not an option at ICML, I will trust the authors and assume they will act in good faith. I am improving my score from "2: Reject" to "4: Borderline Accept" in light of this.

**Paper Summary:**

This paper argues that large-scale ML and AI conferences should incorporate an "LLM Score Calibration" layer where an LLM maps reviewer text into rating scores and flags those reviews with a high discrepancy between the reviewer-assigned scores and the LLM-assigned scores (which only map the reviewer text to score, not the scores which an LLM might assign as an independent reviewer). They propose that this system, if incorporated into the review workflows (such as on OpenReview), should provide an option to the reviewer to either revise their score or revise their text explanation so that the discrepancy is reduced. They state that **unreliable scoring** (score numbers which don't match the review explanation) is one of the most widely blamed source sources for author frustration. They also state that the since the actual score numbers carry an undue weight for the final accept/reject decision, their proposed solution should partially mitigate this pain point.

**Position:**

Yes

**Position In Title:**

Yes

**Related Work:**

3

**Strengths And Weaknesses:**

Strengths:

The topic is clearly of relevance to the ICML (and the broader ML/AI) community and improving peer-review (which this paper targets to do, at least in one aspect) is of great importance to the community. The paper is also likely to inspire discussion. It does cite a decent amount of related work and events, but I cannot authoritatively assess if they missed anything as I do not work in this space (peer review).

Major Weakness:

Although I agree with the spirit of the paper and their general discussion, I have a major issue with their entire "Empirical Evidence" section. In my opinion, anything to do with their LLM scores on the ICLR 2023 and ICLR 2024 data is completely useless due to pretraining-data
contamination. I will address this more thoroughly in the **Questions** section.

Although I would have loved to support this paper (as I think this is genuinely interesting work and also timely), I cannot support it in its current form due to the data-contamination issues with the experiments. If the authors seriously think about these questions and make sure that the experiments are conducted more carefully, this would be a good direction.

**Support:**

1

---

> ### Author Rebuttal · Authors · 2026-03-31
>
> Dear Reviewer,
>
> We sincerely appreciate your view that this is genuinely interesting and timely work and would have loved to support our work. We fully understand your concern about pretraining-data contamination and emphasize that **we did take this issue into consideration in our experiments**. Below are our detailed responses.
>
> > **Q1: Concern about pretraining-data contamination**
>
> **A1:** We took this issue seriously from the beginning, which is why we included a memorization-screening step for ICLR 2023–2024 following two representative studies [1, 2] to **filter out likely memorized papers**. While no screening can guarantee complete detection and removal of contamination, this represents our best-effort attempt based on the available methodology.
>
> [1] Quantifying memorization across neural language models.
>
> [2] Detecting pretraining data from large language models.
>
>
> To further address this concern, we have added experiments beyond the original submission: (1) results on ICLR 2026, NeurIPS 2025 and ICML 2025 data, three clean testbeds with no contamination concerns; (2) rerun of key analyses using an older model whose training strictly predates all review data. Across these experiments, the main trends remain consistent, confirming that observed patterns reflect genuine rationale–score misalignment rather than contamination artifacts. The detailed results are below.
>
> > **Q2: Experiments on ICLR 2023/2024 are useless due to data contamination**
>
> **A2:** We conducted rigorous ablation studies using both un-leaked models and un-leaked datasets. We re-ran experiments for ICLR 2023 and 2024 using `gpt-3.5-turbo-1106`, whose cutoff date strictly precedes the public release of these papers and reviews, making contamination impossible. Results are highly consistent with our original findings:
>
> | Dataset | Bias Span Mean | Mean Residual Std Dev |
> | :--- | :---: | :---: |
> | ICLR 2023 | 2.665 | 1.726 |
> | ICLR 2024 | 2.691 | 1.715 |
>
> We also used `gpt-4o-mini` on ICLR 2026, NeurIPS 2025 and ICML 2025 data—all published after the model's knowledge cutoff, ensuring zero contamination:
>
> | Dataset | Bias Span Mean | Mean Residual Std Dev |
> | :--- | :---: | :---: |
> | ICLR 2026 | 3.601 | 1.643 |
> | ICML 2025 (partial) | 3.950 *(scaled to 1-10)* | 1.826 *(scaled to 1-10)* |
> | NeurIPS 2025 (partial) | 2.980 | 1.337 |
>
> Across all settings, significant bias persists, decisively confirming that inconsistent human rationale-to-score mapping is a robust phenomenon independent of data contamination.
>
> > **Q3: Reason for the ICLR 2025 variance surge**
>
> **A3:** You raised a very reasonable hypothesis that the variance increase in Fig 4 might stem from data shifting from "leaked" to "non-leaked". To isolate this factor, we conducted the same experiment using `gpt-5.4-mini`, whose pre-training explicitly includes ICLR 2023–2025 datasets:
>
> | Year | Mean Variance | Median Variance |
> | :--- | :---: | :---: |
> | **2023** | 0.0830 | 0.0300 |
> | **2024** | 0.0906 | 0.0400 |
> | **2025** | 0.3496 | 0.2755 |
>
> Even with the model having "memorized" all three years, ICLR 2025 variance remains significantly higher than 2023 and 2024. This indicates the increase does not stem from leakage, but likely from the surge in submissions leading to greater quality heterogeneity and inconsistent scoring.
>
> > **Q4: Citation analysis is useless due to leakage**
>
> **A4:** To address the concern of potential citation data leakage, we re-conducted borderline paper citation analysis using the un-leaked model `gpt-3.5-turbo-1106` on ICLR 2023/2024 (training cutoff predates both publication and citation accumulation):
>
> | Year | Score Band | Reviewer Mean Cit. | Anchor Mean Cit. | Improvement |
> | :--- | :---: | :---: | :---: | :---: |
> | 2023 | 4–5 | 13.61 | 18.07 | +32.8% |
> | 2023 | 5–6 | 22.67 | 43.35 | +91.2% |
> | 2024 | 4–5 | 13.10 | 19.49 | +48.8% |
> | 2024 | 5–6 | 29.06 | 32.20 | +10.8% |
>
> We also ran citation analysis on ICLR 2025 using `gpt-4o-mini`, where leakage is physically impossible:
>
> | Score Band | Reviewer Mean Cit. | Anchor Mean Cit. | Improvement |
> | :---: | :---: | :---: | :---: |
> | 4–5 | 12.53 | 14.02 | +11.9% |
> | 5–6 | 15.14 | 20.40 | +34.8% |
>
> Across all settings, Anchor Scores consistently identify higher-impact borderline papers. Moreover, our method converts human textual opinions into scores rather than predicting citations directly, making it inherently resistant to citation leakage.
>
> Once again, thank you for pushing us toward rigorous experimental validation. We hope these experiments fully address your concerns and demonstrate the robust value of our proposed system.
>
> ***

---

> > ### Author Rebuttal · Reviewer_v6RF · 2026-04-04
> >
> > The authors have re-run several key experiments in their paper to address the main issue of data-contamination that I had raised. Ideally, I would have liked to see them incorporate all of these observations in the main paper and revise everything before giving the go-ahead but since that is not an option at ICML, I will trust the authors and assume they will act in good faith. I am improving my score from "2: Reject" to "4: Borderline Accept" in light of this.

---

### Decision · Program_Chairs · 2026-04-30

**Decision:**

Accept (regular)

**Comment:**

Reviewers point to the specific topic of peer review as being a primary strength of the position. Unsurprisingly, it is considered timely in the context of its own review, but will remain timely given the growing burden of peer review at ICML and elsewhere. While reviewers overall found the proposal compelling, there were several substantive concerns raised during review, including the lack of human annotation as a form of validation and concerns that using past conference may display leakage with respect to LLM training. These concerns are partially addressed in rebuttal, though there is acknowledgement that a lot is left to text / results that will be included that were not available in the initial review. Overall, this could be an interesting paper for ICML as there is generally a great deal of interest in peer review as a topic.